# Lensfree OLEDs with over 50% external quantum efficiency via external scattering and horizontally oriented emitters201895

Jinouk Song[1], Kwon-Hyeon Kim [2], Eunhye Kim[1], Chang-Ki Moon[2], Yun-Hi Kim[3], Jang-Joo Kim[2] & Seunghyup Yoo[1]

High efficiency is important for successful deployment of any light sources. Continued efforts have recently made it possible to demonstrate organic light-emitting diodes with efficiency comparable to that of inorganic light-emitting diodes. However, such achievements were possible only with the help of a macroscopic lens or complex internal nanostructures, both of which undermine the key benefits of organic light-emitting diodes as an affordable planar light source. Here we present a systematic way to achieve organic light-emitting diodes with ultrahigh efficiency even only with an external scattering film, one of the simplest low-cost outcoupling structures. Through a global, multivariable analysis, we show that scattering with a high degree of forwardness has a potential to play a critical role in realizing ultimate efficiency. Combined with horizontally oriented emitters, organic light-emitting diodes equipped with particle-embedded films tailored for forward-intensive scattering achieve a maximum external quantum efficiency of 56%.

[1] School of Electrical Engineering, Korea Advanced Institute of Science and Technology (KAIST), Daejeon 34141, Republic of Korea. [2] Department of Materials Science and Engineering, Seoul National University, Seoul 151-744, Republic of Korea. [3] Department of Chemistry and Engineering Research Institute (ERI), Gyeongsang National University, Jinju 66-701, Republic of Korea. Correspondence and requests for materials should be addressed to J.-J.K. (email: jjkim@snu.ac.kr) or to S.Y. (email: syoo@ee.kaist.ac.kr)

The development of high-efficiency light sources is important for many aspects of modern life, from the increasing demand for longer operation time in personal portable devices to the worldwide need to reduce energy consumption. While organic light-emitting diodes (OLEDs) have been regarded as an ideal light source for both displays and lighting, their efficiency has been lagging behind that of III−V compound semiconductor LEDs. Fortunately, with the introduction of phosphorescent emitters or those based on thermally activated delayed fluorescence, internal quantum efficiency of OLEDs has become close to unity[1–7], serving as a firm foundation for ultra-efficient OLEDs. Nevertheless, the efficiency of most OLEDs is still limited to the range of 20 to 30% mainly due to their limited outcoupling efficiency stemming from the total internal reflection occurring at interfaces with different refractive indices, absorption, and loss to surface plasmon polariton modes at metal/organic interfaces. For this reason, numerous light outcoupling schemes have been proposed for OLEDs and have proven effective in increasing their external quantum efficiency (EQE)[8–15]. These light outcoupling schemes may be categorized as internal or external depending on whether the structural modification is made near/inside the active layers or at the surface of the substrates away from the active layers, respectively. Although combining both internal and external outcoupling schemes may result in the highest possible efficiency, internal outcoupling schemes can often be subject to electrical short, local high-field-induced degradation, limited scalability of manufacturing, and/or increased fabrication cost. Likewise, although a half-ball lens has often been used as an external outcoupling structure and proven so effective in light extraction that OLEDs with EQE of 50% or higher can be realized with it[13,16], it is intended mainly for demonstration, rather than for any practical purpose; its use is limited to small-scale devices from the practical point of view and, furthermore, contradicts with a key benefit of an OLED as a planar or flexible light source. Therefore, it would be highly beneficial to develop a systematic way to significantly enhance outcoupling efficiency solely with scalable external outcoupling structures, such as microlens array foil, substrate texturing, or bulk-scattering film.

Among those external outcoupling schemes, we focus on a bulk-scattering-based approach because it meets most of the requirements for ideal light extraction schemes. In addition to its effectiveness for outcoupling enhancement, it has the following advantages: it can maintain, unlike a half-ball lens, the planar geometry and compatibility with flexibility; it is easy to fabricate on a large scale at low cost; and it can reduce the angular color shift through the inherent stochastic nature of the scattering processes. While previous works mostly focused on the realization of a scattering film itself and on demonstration of its feasibility[17–19], in this work, we try to systematically maximize its light extraction effect so that very high-efficiency OLEDs can be achieved even with external scattering films. To this end, we first generalize the bulk or volumetric scattering phenomena within the framework of radiative transfer theory (RTT)[20], which treats a scattering film as an infinite homogeneous layer and describes a change in optical flux along a given direction due to absorption, scattering, and emission (if any) that can occur during light propagation. Unlike the Monte-Carlo approach, in which individual scattering events and successive light propagations are traced for every step[21], RTT takes advantages of the statistical nature of multiple scattering processes and describes their average effect on intensity change via a well-defined set of equations[22]. This equation-based character of the RTT makes it easy to combine, in a trans-scale fashion, with an optical model for light emission within an OLED, which may be generalized as radiative emission from a dipole in a thin-film multilayer stack. This then enables global, multivariable analysis at high speed, in which an OLED with an external scattering layer is considered as a whole[23], rather than as two separate entities, so the design parameters of the scattering layer can be determined for maximal EQE in conjunction with those of the OLED layer configuration.

With this approach, we here demonstrate that optimized external scattering layers, together with highly oriented dipole emitters, are able to yield OLEDs having EQE greater than 50% even without use of a macroscopic lens or complex internal nanostructure. Tailoring the characteristics of particle-embedded scattering layers such as asymmetry parameter, scattering efficiency, and scatterance is shown to play a key role in utilizing the full potential offered by the external scattering methods.

## Results

**Global analysis of an OLED with an external scattering film.** The calculation process in the proposed optical simulation is schematically presented in Fig. 1a. First, the radiant intensity of light with the wavelength of $\lambda$ and angle $\theta$ exiting into a scattering layer (SL) $[=I_{SL}(\theta,\lambda)]$ is calculated using the coherent dipole radiation theory following the formalism summarized by Neyts[24] and Moon et al.[25]. The following parameters of the OLED stack influence $I_{SL}(\theta,\lambda)$: the refractive indices ($n_j$) and thicknesses ($d_j$) of the constituent layers with "$j$" being the index designated for each layer, and the horizontal dipole ratio ($\Theta$) and intrinsic radiative quantum efficiency ($q$) of an emitter. For a scattering layer of interest having the host refractive index of $n_{SL}$, Mie theory is used for embedded scattering particles (SPs) having the concentration, refractive index, and diameter of $\rho_{SP}$, $n_{SP}$, and $d_{SP}$, respectively, to obtain the key scattering parameters such as asymmetry parameter ($g$) and scatterance ($S$) of a scattering medium made therewith. The asymmetry parameter $g$ is defined as $<\cos\theta_{sc}>$ for the deflection angle ($\theta_{sc}$) per scattering event, and $S$ quantifies the fraction of the radiant flux lost due to scattering and is defined as $\ln(1/T_{inline})$ for inline transmittance ($T_{inline}$) through a given scattering medium, being analogous to absorbance in absorption[26] (see Supplementary Fig. 1 and description therein for further details). Reflectance of an OLED $(=R_{OLED}(\theta,\lambda))$ for light returning into the OLED stack from the scattering layer is calculated based on thin-film optics. A seamless trans-scale calculation is then enabled by employing $R_{OLED}(\theta,\lambda)$ as a boundary condition for the scattering layer at its interface with the OLED and by entering $I_{SL}(\theta,\lambda)$ as an initial value for RTT calculation within the scattering layer. In this way, the final angular intensity $(=I_{Air}(\theta,\lambda))$ as well as the EQE $(=\eta_{EQE})$ of the whole device is obtained, potentially as a function of all of the parameters mentioned above.

The results of global analysis performed to design an OLED with the highest possible EQE are presented as a function of active layer thickness, which is the sum of indium tin oxide (ITO) and organic layer thicknesses ($d_{active} \equiv d_{ITO} + d_{org.}$), of the OLED and $g$ and $S$ of a scattering layer in Fig. 1b−e. The detailed OLED thin film structure and the optical constants of the layers used in this work are shown in Fig. 1a and Supplementary Fig. 2, respectively. Each data point in Fig. 1b was obtained by identifying a combination of $d_{ITO}$ and the thicknesses of electron and hole transport layers ($d_{ETL}$, $d_{HTL}$) leading to the maximum achievable EQE ($\eta_{EQE}^{(max)}$), for a given set of {$d_{active}$, $g$, $S$}, as shown in Fig. 1c. In the simulation, $d_{ITO}$ was chosen to be no smaller than 50 nm to ensure the sheet resistance of ITO is not too large.

While a high degree of horizontal dipole orientation is known to be critical in achieving maximal outcoupling in planar OLEDs without an outcoupling structure[27–32], it is herein revealed that this is still the case even in the presence of an external scattering

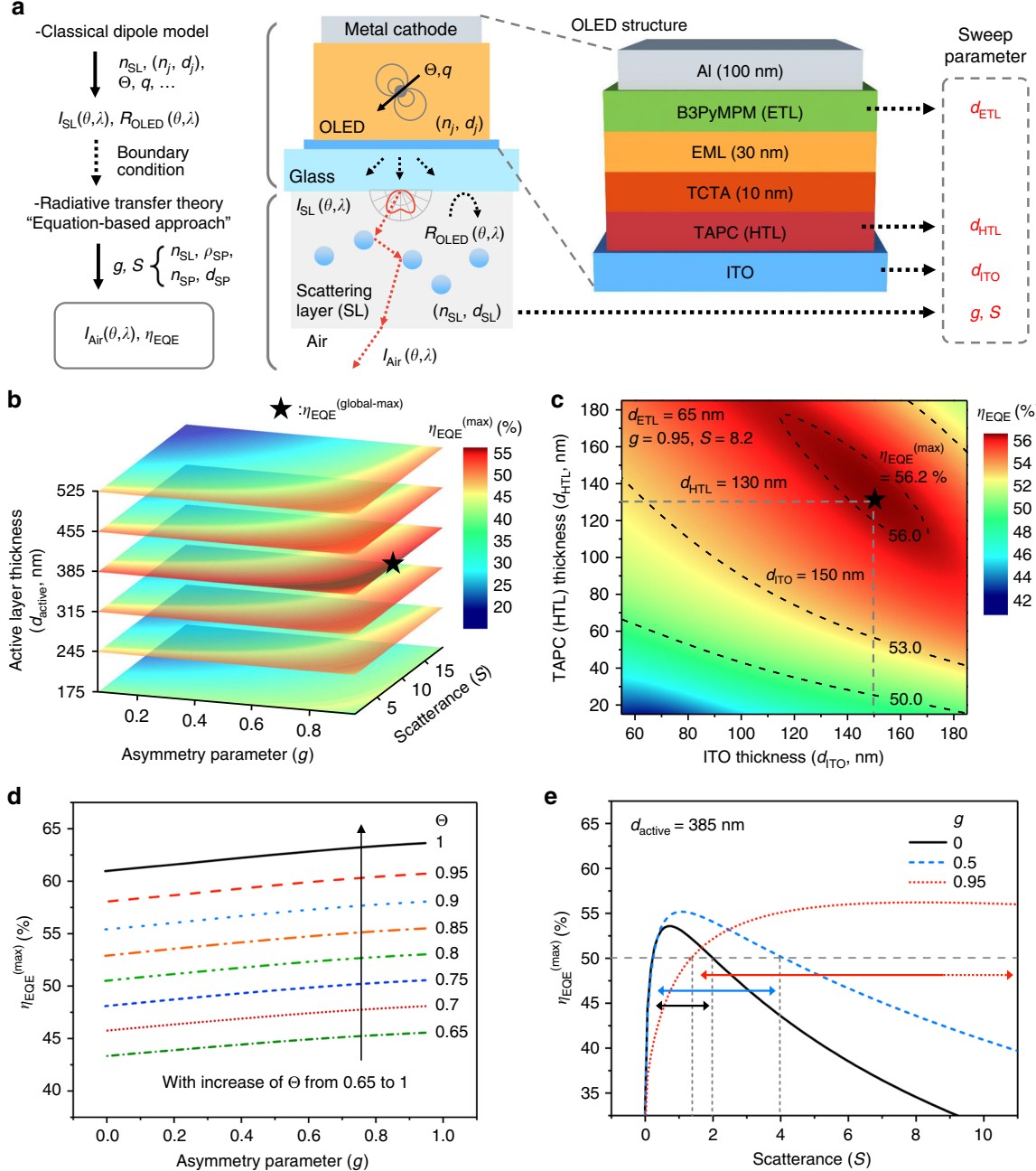

**Fig. 1** Schematic diagram of the proposed trans-scale optical simulation and global optimization results. **a** Schematic illustration of suggested trans-scale optical simulation and schematic diagram of the detailed organic light-emitting diode (OLED) structure under study as well as parameters used in multivariable analysis. **b** Three-dimensional slice plot of maximum achievable external quantum efficiency (EQE) [$=\eta_{EQE}^{(max)}$] vs. active layer thickness ($d_{active} = d_{ITO} + d_{org.}$), asymmetry parameter ($g$), and scatterance ($S$). $d_{ITO}$, $d_{ETL}$, and $d_{HTL}$ were varied to find $\eta_{EQE}^{(max)}$ for each $\{d_{active}, g, S\}$. Here, $d_x$ denotes the thickness of layer "$x$" while "org.", ITO, ETL, and HTL refer to organic, indium tin oxide, electron transport layer, and hole transport layer, respectively. The global maximum considering all the parameters is represented by a black star. **c** The contour plot of EQE vs. $d_{ITO}$ and $d_{HTL}$ for $d_{ETL} = 65$ nm, $g = 0.95$, and $S = 8.2$, which correspond to the global maximum of $\eta_{EQE}^{(max)}$. **d** The $\eta_{EQE}^{(max)}$ of the given structure obtained with variation of the horizontal dipole ratio ($\Theta$) and $g$. **e** The $\eta_{EQE}^{(max)}$ as a function of $S$ in the cases of $g = 0$, 0.5, and 0.95 when $d_{active} = 385$ nm. Color-matched horizontal arrows indicate the ranges of $S$ where $\eta_{EQE}^{(max)}$ is larger than 50% for each graph

layer (see Fig. 1d). That is, $\eta_{EQE}^{(max)}$ for an OLED with an external scattering layer also increases with $\Theta$. The difference in $\eta_{EQE}^{(max)}$ made with $\Theta$ is significant; $\eta_{EQE}^{(max)}$ of an OLED with all of its emitters oriented horizontally ($\Theta = 1$) can reach 63%, while it is limited to 45% for an OLED with isotropically oriented emitters ($\Theta = 0.67$). In the calculations shown in Fig. 1b–e and experiments that follow, therefore, we employed a

phosphorescent emitter of Ir(dmppy-ph)$_2$tmd proposed by Kim et al. to take full advantage of its highly horizontally oriented dipole orientation ($\Theta = 0.865$) as well as its high intrinsic radiative quantum efficiency ($q = 0.98$)[33].

In the proposed device configuration, the trans-scale simulation shows that $\eta_{EQE}^{(max)}$ can reach 56.2% when $S$, $d_{ITO}$, $d_{ETL}$, and $d_{HTL}$ are 8.2, 150 nm, 65 nm, and 130 nm, respectively, with $g$

approaching unity. In this active layer configuration, an OLED without a scattering layer would have $\eta_{EQE}$ of ca. 28 % (Supplementary Fig. 3a, b). If an OLED without a scattering layer took the same configuration, but the emitter is randomly oriented ($\Theta = 0.67$), $\eta_{EQE}$ would be limited at ca. 23 % (Supplementary Fig. 3c, d). The horizontal dipole orientation and the optimized scattering layer in the OLED under study are thus responsible for the EQE enhancement ratio of 1.2 (=28/23) and 2.0 (=56/28), respectively. The ratio of combined enhancement is therefore 2.4 (=1.2×2.0) when compared with the EQE of the scattering-layer-free OLED based on an emitter having the same emission spectra and $q$ as Ir(dmppy-ph)$_2$tmd but having $\Theta$ of 0.67.

Two important aspects can be noted from the overall trend in Fig. 1b. First, the dependence of $\eta_{EQE}^{(max)}$ on $d_{active}$ for a given set of $g$ and $S$ is relatively mild but still significant; therefore, it is important to identify the optimal active layer thickness to realize the maximal efficiency with scattering-based extraction. Power dissipation spectra obtained for the base reference device structure indicate that the optimal case ($d_{active} = 385$ nm) is close to the case where the sum of air modes and substrate modes is at its maximum, rather than the case where only the air mode is maximized (see Supplementary Fig. 3). Second, it is more advantageous to have $g$ close to unity than to have $g$ below 0.5; $\eta_{EQE}^{(max)}$ can be higher and ultimately larger than 50% for a wide range of $S$ (see Fig. 1e). For example, in the case of near-isotropic scattering, that is, when $g$ becomes close to zero, $\eta_{EQE}^{(max)}$ is about 50% only with $S$ between 0.2 and 2. However, when there is forward-intensive scattering, that is, when $g$ becomes close to unity, this can result in the same level of $\eta_{EQE}^{(max)}$ in a wide range of $S$ covering from 1.5 all the way to ca. 10 (or beyond), with a fairly flat $\eta_{EQE}^{(max)}$ of ca. 55−56% for $S > 4$. This can be regarded practically useful not only in that it allows us to tap into the highest efficiency a given system can offer but also in that it does so with a wide fabrication margin. The almost flat $\eta_{EQE}^{(max)}$ over a wide range of $S$ will also be useful in maintaining panel-to-panel brightness uniformity when OLED lighting panels are used in a modular, matrix configuration to light a room. In the case of moderate forward scattering, the trend of $\eta_{EQE}^{(max)}$ over $S$ evolves from that of the isotropic scattering to that of forward-intensive scattering as $g$ increases. When $g = 0.5$, for example, $\eta_{EQE}^{(max)}$ can ultimately be as high as 55.1% and can be higher than 50% with $S$ between 0.2 and 4. Note that this optimal range of $S$ is wider than that of the isotropic scattering but narrower than that of the forward-intensive scattering.

**Design and characterization of scattering films**. To investigate the condition for high $g$, Mie theory was used to calculate $g$ for spherical dielectric particles with the refractive index and diameter of $n_{SP}$ and $d_{SP}$, respectively, which are embedded in a host medium. The refractive index of the host medium ($n_{SL}$) was fixed at 1.57, which is the refractive index of the UV-curable resin (NOA73, Norland Products, Inc.) used in this work. In addition to $g$, we calculated scattering efficiency ($Q_{sc}$), which is defined as the ratio of the effective scattering cross-section ($\sigma_{sc}$) to the geometrical cross-section of a spherical particle ($\pi d_{SP}^2/4$). $Q_{sc}$ is regarded important as it indicates how effective a scattering process is for a given combination of a host medium and a scatterer. From the practical point of view, $Q_{sc}/d_{SP}$ directly scales with the upper limit for the scattering coefficient ($\mu_{sc}$) or the inverse of the mean free path ($=1/L_{MFP}$) (Supplementary Fig. 4 and description therein for further details).

As shown in Fig. 2a, calculation done at the wavelength ($\lambda$) of 532 nm reveals that a high degree of forward scattering can be achieved when $d_{SP}$ is comparable to $\lambda$ or higher and $|n_{SP} - n_{SL}|$ is

not too large. For example, $g$ can be larger than 0.9 when $d_{SP} >$ ca. 500 nm and $\Delta n \equiv |n_{SP} - n_{SL}| < $ ca. 0.25. Note that $g$ approaches the unity as $\Delta n$ further gets reduced. However, one has to take caution not to make $\Delta n$ too small because it can lead to too low $Q_{sc}$ (see Fig. 2b). In such a case, the minimum achievable $L_{MFP}$ could become so large that $S$ of 1 or higher may not be obtained with a practical range of the thickness of a scattering layer ($d_{SL}$). For example, when $n_{SP} = 1.56$ ($\Delta n = 0.01$) for $n_{SL} = 1.57$ and $d_{SP} = 1$ μm, $Q_{sc}$ becomes as low as $6.7 \times 10^{-3}$; in this case, the minimum $L_{MFP}$ achievable with a simple cubic arrangement ($=L_{MFP}^{(c,min)}$) becomes approximately 0.2 mm. Because $S = \ln(1/T_{inline}) = d_{SL}/L_{MFP}$, this means that at least 0.2-mm-thick scattering layer is required to achieve $S$ of 1. Hence, achieving $S$ of 5−10 in such a case will require $d_{SL}$ of at least 1−2 mm. Note that these minimum $d_{SL}$ values are for closely packed cases; for most practical range of concentrations, the actual layer thickness required to achieve $S$ of 5−10 is likely to be too large to be considered practical when $\Delta n = 0.01$.

In consideration of both $g$ and $Q_{sc}$, therefore, scattering layers were fabricated in this work by dispersing silicon dioxide (SiO$_2$) spherical particles ($n_{SP} = 1.46$) with $d_{SP}$ of 800 nm in the NOA73 resin. In this condition, $g$ can become higher than 0.9 yet $Q_{sc}$ can still be around 0.45; in this case, $L_{MFP}^{(c,min)}$ is as small as 1.8 μm, making it plausible to realize $S$ of 5−10 with $d_{SL}$ less than a few hundreds of μm even with a reasonably dilute condition that can be easily achieved in practice. To vary $S$, concentrations of 0.2, 0.5, 1, 1.5, 3, and 4.5 wt.% were tried with $d_{SL}$ fixed at approximately 290 μm (see Methods section). A scanning electron microscope image of a fabricated SiO$_2$-based scattering layer as well as a representative photograph of a scattering layer obtained from 3 wt.% solution is shown in Fig. 2c, d. The angular intensities of the light transmitted through these scattering films shown in Fig. 2e indicate that the measured data can be fitted well over a wide range of particle concentrations with simulation curves obtained for $g = 0.91 \pm 0.01$ using Monte-Carlo simulation done with LightTools[TM][21]. This is consistent with the Mie theory result, which predicted a $g$ value of 0.95. For comparison's sake, scattering layers based on TiO$_2$ nanoparticles with $n_{SP}$ of ca. 2.5 and $d_{SP}$ of about 200 nm (SG-TO200, Sukgyung AT Co., Ltd.) were also realized with the concentrations of 0.05, 0.1, 0.2, 0.5, 1, and 1.5 wt.%. These TiO$_2$-based scattering layers exhibited $g$ of $0.68 \pm 0.03$, which falls on moderate forward scattering and is consistent with the calculation results shown in Fig. 2a (Supplementary Fig. 5 for the experimental results of TiO$_2$-based samples). $Q_{sc}$ in this case was calculated to be as high as 2.2, indicating that the $L_{MFP}^{(c,min)}$ can be on the order of 0.1 μm. Similarly, air voids ($n = 1$)[19] may also serve as useful scatterers; calculation indicates that $g$ of up to ca. 0.8 and $Q_{sc}$ approximately 1.8 can be readily available when used with the host medium in this work ($\Delta n = 0.57$) provided that $d_{SP}$ is greater than ca. 400 nm. In both TiO$_2$ and air void cases, however, care must be taken when choosing $d_{SP}$ because $g$ values tend to exhibit an oscillatory behavior over $d_{SP}$ with a system having a relatively large $\Delta n$ as shown in Fig. 2a.

**Performance and analysis of fabricated OLEDs**. Figure 3 presents the optoelectrical characteristics of the devices fabricated with the SiO$_2$-embedded scattering films with the thicknesses of the ITO and organic layers set at the optimal values obtained for the global maximum EQE. As an expected benefit of the external outcoupling scheme rather than any internal micro- or nano-structuring, not only current density features virtually no or very low leakage (on the order of nA cm$^{-2}$) even near the turn-on voltage (=ca. 2.4 V), but also current density−voltage curves are highly reproducible (see Fig. 3a). This allows for reliable

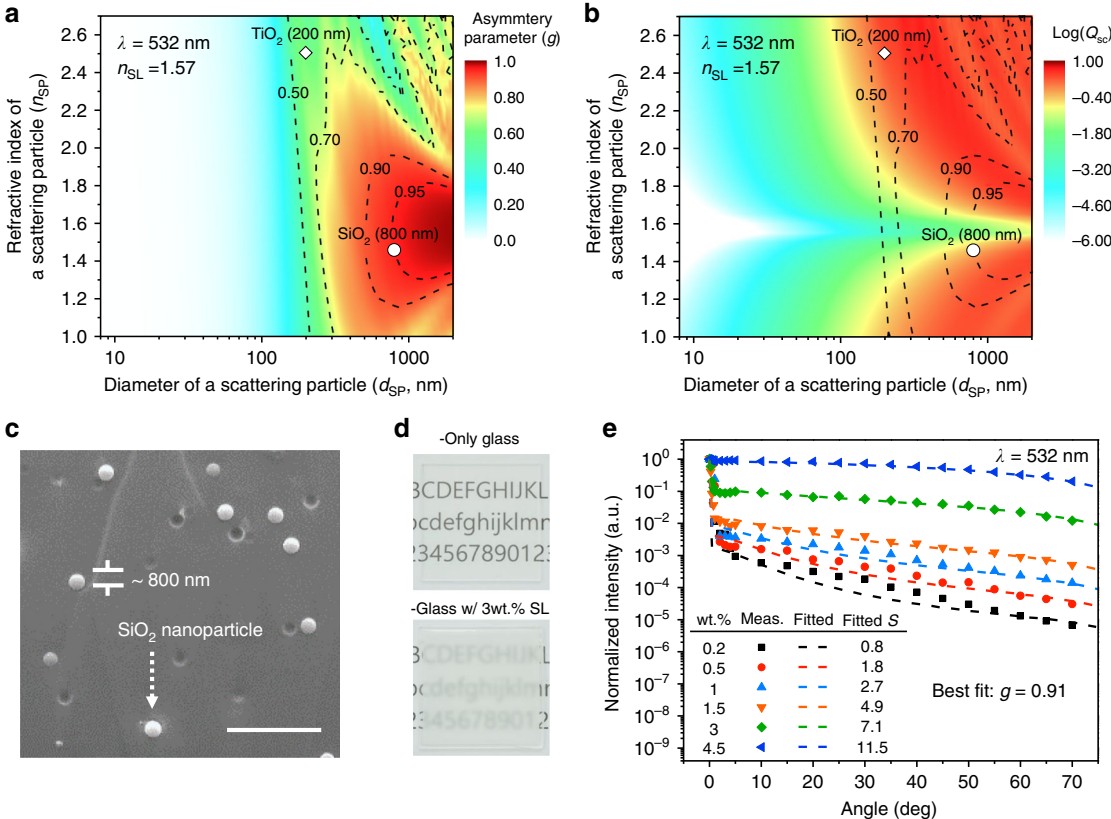

**Fig. 2** Design and analysis of a scattering layer. **a, b** Calculated (**a**) asymmetry parameter ($g$) and (**b**) logarithm of the scattering efficiency ($Q_{sc}$) of a scattering layer at the wavelength ($\lambda$) of 532 nm according to the diameter ($d_{SP}$) and refractive index ($n_{SP}$) of a scattering particle (SP) in the host (NOA 73, $n_{SL} \sim 1.57$). **c** A scanning electron microscope image of a fabricated $SiO_2$-based scattering layer (scale bar: 5 μm). **d** Photographs of glass without a scattering layer and with a scattering layer (SL) fabricated using 3 wt.% solution. **e** Measured (dot) and fitted (dashed) graphs of angular characteristic of scattered light after passing through $SiO_2$ embedded scattering layers. 532 nm-laser was used for the measurement

measurement of EQE and power efficiency even at low brightness levels, which is important in properly comparing the max EQE values from the simulated and experimental results[34]. As can be seen in Table 1 and Fig. 3b−d, the maximum EQE, obtained with the $SiO_2$-based scattering layer having $g$ and $S$ about 0.91 and 7.1, respectively, was as large as 50.9%, which corresponded to a power efficiency of 197 lm W$^{-1}$ and a current efficiency of 167 cd A$^{-1}$. Photographs of $SiO_2$-based OLEDs operating at current density of 0.1 mA cm$^{-2}$ are shown in Fig. 3e.

To confirm the reliability of the proposed trans-scale optical simulation, the predicted characteristics from simulation are compared with the measured device performance (see Fig. 4). The EQE at 10 cd m$^{-2}$ in relation to $S$ is presented in Fig. 4a; in the case of OLEDs with the $SiO_2$-based scattering layers, it exhibited good matching with the trend predicted through the proposed trans-scale simulation done with $g$ given as 0.91. The EQE increased rapidly with $S$ but tended to increase only slightly above a certain level of $S$, which in this case was about 4. In contrast, the EQE of OLEDs with the $TiO_2$-based scattering layers ($g = 0.68$) peaked at a relatively low $S$ (~2) and then decreased rather rapidly as $S$ further increased, making the range of $S$ leading to EQE > 50% narrower than the devices with the $SiO_2$-based scattering layers. Nevertheless, OLEDs with the $TiO_2$-based scattering layers also exhibited very high EQE up to 52.0% and thus should be useful in many applications, provided that the thickness and concentration of the scattering layers are tightly controlled so that $S$ may be at its optimum. As the optimal $S$ is relatively low as shown in Figs. 1d and 4a, $TiO_2$-based scattering scheme will

particularly be useful in highly flexible OLEDs, which can benefit from ultrathin substrates. Its high $Q_{sc}$ (~2.2) that can lead to the relatively short $L_{MFP}$ can also be helpful in this respect.

While the results obtained from both types of OLEDs exhibited EQE greater than 50% within their respective optimal $S$ ranges in Fig. 4a, the measured EQE values were off from the values predicted through the trans-scale RTT simulation by approx. 5% for $SiO_2$-based devices and 3% for the $TiO_2$-based devices; this discrepancy can be attributed mainly to the fact that the simulation assumed infinitely large OLEDs, while actual devices have a finite spatial extent. It is noteworthy that such a difference is more pronounced in the $SiO_2$-based devices. This is because a high-$g$ scattering system tends to yield only a small deflection per each scattering event, and thus the confined light therein often travels a longer lateral distance before being extracted than in a low-$g$ scattering system. When a CAD-based commercial software (LightTools$^{TM}$) was used to reflect the real dimensions and layout of the fabricated OLEDs, the simulated results (the dashed lines in Fig. 4a) showed a good agreement with the experimental data (see Supplementary Fig. 6 for simulation details and Supplementary Fig. 7 for the performance of OLEDs with $TiO_2$-embedded scattering layers). This issue would become marginal in OLEDs used in practical lighting applications, which typically have a device area much larger than several cm². In experiments done with OLEDs having a larger substrate (40 mm × 70 mm) and back reflector (38 mm × 38 mm) (=Type B) as opposed to the geometry used for the OLEDs in EQE−$S$ trend experiment (=Type A; 25 mm × 25 mm substrate with 23 mm ×

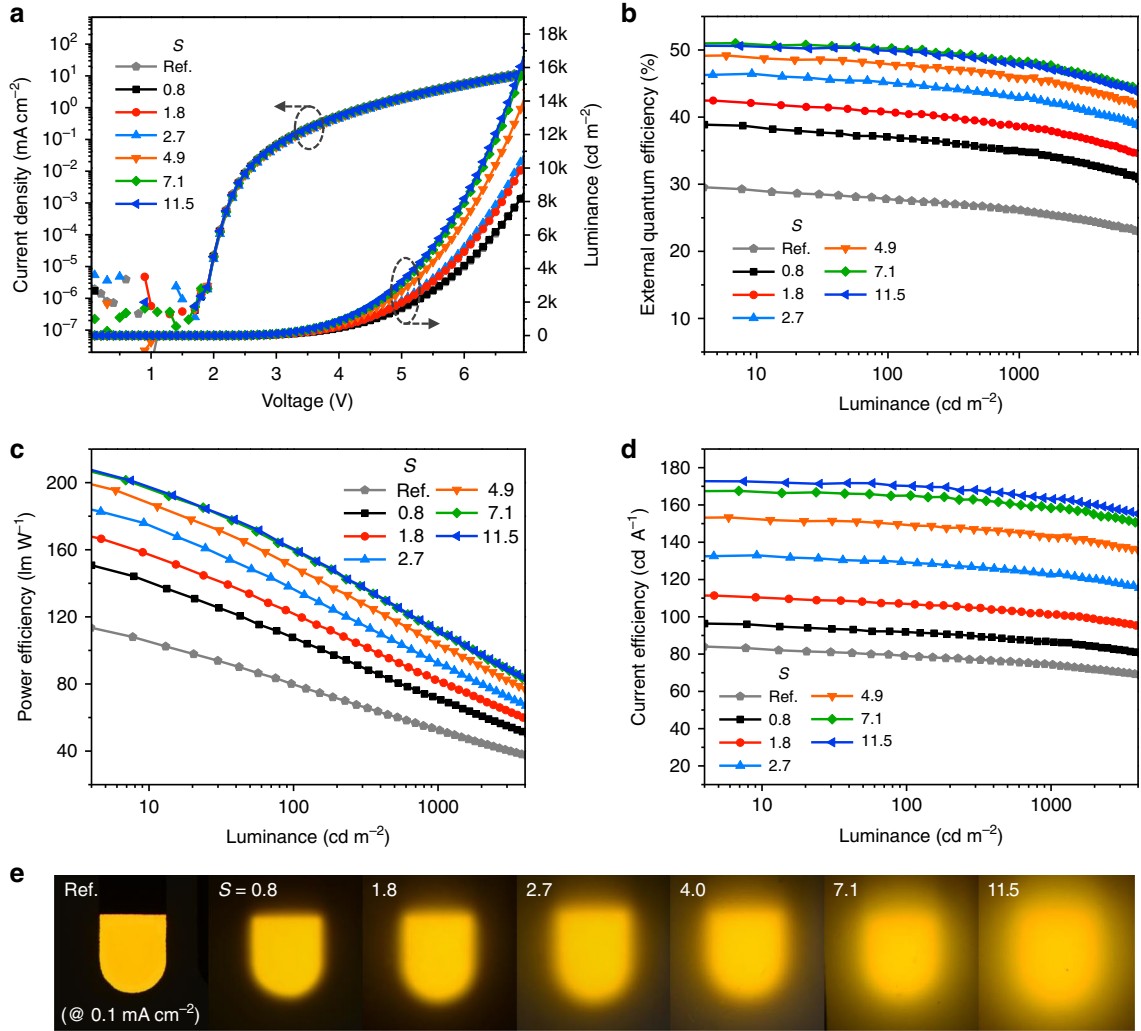

**Fig. 3** Characteristics of organic light-emitting diodes (OLEDs) with SiO$_2$-based scattering layers. **a** Current density ($J$)−luminance ($L$)−voltage ($V$) characteristics. **b**−**d** External quantum efficiency (**b**), power efficiency (**c**), and current efficiency (**d**) vs. $L$. Layout of the devices used in this figure follows that of Type A defined in Fig. 4b. **e** Photographs of OLEDs with SiO$_2$-embedded scattering layers according to scatterance ($S$). Photographs were taken at current density of 0.1 mA cm$^{-2}$

11 mm reflector) (Fig. 4b for the detailed device layout), the EQE of an OLED with the SiO$_2$-based scattering layer ($g = 0.91$, $S = 7.1$) and that with the TiO$_2$-based scattering layer ($g = 0.68$, $S = 2.0$) were measured to be as high as 56.3 and 54.5%, respectively, as shown in Fig. 4a (the hollow circle and triangle), and Fig. 4c−e. These values are close to the respective global maxima (55.9 and 55.6% for $g = 0.91$ and $g = 0.68$, respectively) predicted by the proposed RTT simulation assuming the infinitely large area. The corresponding power efficiencies of these devices at 10 cd m$^{-2}$ were 221 and 211 lm W$^{-1}$, respectively. From the photographs shown in Fig. 4b, it can be confirmed that the distribution of light at the exit surface of the OLED with the SiO$_2$-based scattering layer does have a larger spatial extent than that from the OLED with the TiO$_2$-based scattering layer.

In achieving such high EQEs, one cannot overemphasize the importance of using highly horizontally oriented dipole emitters together with the proposed scattering layers. In a comparisonal experiment done with Ir(ppy)$_2$acac and Ir(ppy)$_3$—widely studied phosphorescent emitters with $\Theta$ of 0.76 (i.e. slight preference toward horizontal dipole orientation) and 0.67 (i.e. random, isotropic dipole orientation)[35], respectively, optimized OLED devices (Type A) coupled with the SiO$_2$-based scattering layers exhibited a similar trend for EQE vs. $S$ with the maximum EQE as high as 44.8 and 40.3%, respectively (Supplementary Figs. 8, 9 and description provided therein for further details).

Another critical benefit of having large $g$ is well illustrated in Fig. 5a, b. The fact that $\eta_{EQE}^{(max)}$ is obtained with a relatively large $S$ is advantageous in that efficient devices can take full advantage of the randomizing effect of scattering that can effectively suppress the angular color shift of the emitted light. It can be clearly seen in Fig. 5b that OLED devices with a scattering layer could still undergo a nonnegligible angular color shift if the scattering layer has a relatively low $S$ of around 1 to 2. In contrast, the proposed OLEDs with $S$ of about 7 exhibit color-stable performance regardless of the observation angle with ideal Lambertian-like angular characteristics (see Supplementary Fig. 10).

While the analysis and experiment in this work were done in the limited spectral range that belongs to green and orange, the proposed approach will be likely to work well for other colors due to the relatively color neutral characteristics of Mie scattering and due to the fact that many of the glassy materials tend to exhibit a

**Table 1 Summary of the performance of the OLEDs with SiO$_2$-based scattering layers having various $S$**

| $S$ | External quantum efficiency (%) | | Power efficiency (lm W$^{-1}$) | | Current efficiency (cd A$^{-1}$) | |
|---|---|---|---|---|---|---|
| | at $L = 10^a$ | 1000 | 10 | 1000 | 10 | 1000 |
| Ref. | 29.0 | 26.1 | 105.1 | 52.5 | 82.7 | 74.3 |
| 0.8 | 38.5 | 34.9 | 140.7 | 71.0 | 95.5 | 86.6 |
| 1.8 | 42.1 | 38.6 | 157.5 | 81.9 | 110.4 | 101.1 |
| 2.7 | 46.5 | 43.0 | 174.9 | 92.0 | 133.2 | 123.0 |
| 4.9 | 49.0 | 45.8 | 188.6 | 103.6 | 152.7 | 142.9 |
| 7.1 | 50.9 | 48.1 | 196.8 | 111.6 | 167.0 | 158.1 |
| | (56.3$^b$) | (52.6) | (221.1) | (136.4) | (169.6) | (158.4) |
| 11.5 | 50.6 | 47.8 | 198.4 | 112.3 | 172.8 | 163.2 |

$^a$Obtained at indicated luminance ($L$) (unit: cd m$^{-2}$)
$^b$The values inside parenthesis are for the Type-B OLED in Fig. 4. All the other data are from Type-A OLEDs

similar trend in their dispersion relations; i.e. individual $n_{SP}$ and $n_{SL}$ do vary with $\lambda$, but both of them tend to increase as $\lambda$ gets smaller, making $\Delta n$ remain almost unchanged in many cases. On the other hand, the ratio of $d_{SP}$ to $\lambda$ becomes smaller as $\lambda$ increases, and thus $g$ tends to decrease with $\lambda$. Nevertheless, the change remains minor in the visible spectral range (see Supplementary Fig. 11).

## Discussion

In summary, we explored a systematic way to achieve an EQE of 50% or higher in OLEDs that employ only an external scattering medium rather than resorting to a macroscopic lens or complex internal structuring. A trans-scale simulation was carried out that combined the RTT for scattering films and the power-dissipation model for light-emission in multilayer thin films. Thanks to the equation-based nature of RTT, a facile global optimization could be done while simultaneously taking into account several variables for OLED stacks and scattering layers. Through this study, we identified the following two important aspects: first, a scattering medium with a high asymmetry parameter ($g$), achievable with the case of low $\Delta n$ and relatively large $d_{SP}$, is essential in achieving the ultimately high $\eta_{EQE}^{(max)}$ over a wide range of $S$; second, a high degree of horizontal orientation of dipole emitters is still important even when used together with an external scattering medium. It was further identified that a scattering medium with a moderate degree of $g$ can also lead to high $\eta_{EQE}^{(max)}$ provided $S$ is tightly controlled at near the optimum considering its relatively narrow range of $S$ leading to high efficiency. Nevertheless, the moderate-$g$ scattering system has a relatively small optimal $S$ and can be realized with a large $\Delta n$ that can result in high $Q_{sc}$. These characteristics are expected to be useful in applications where a highly thin form factor is important (e.g. ultraflexible OLEDs), because high efficiency can be achieved with a relatively thin scattering layer in this case.

Coupled with nanoparticle-based scattering layers tailored via Mie theory for high $g$ and reasonable range of scattering efficiency ($Q_{sc}$), optimized OLED stacks based on highly horizontally oriented dipole emitters exhibited a maximum EQE of 56% and a power efficiency of 221 lm W$^{-1}$. This result is a significant achievement in that such high efficiency is demonstrated without a half-ball lens or a complex internal outcoupling scheme. Given the simplicity of preparing scattering films and their immense benefits such as angular color shift suppression, the proposed approach can provide OLEDs with a practically viable route to ultrahigh efficiency that was available only to state-of-the-art III−V LEDs, while keeping all the form factor advantages that organic technologies can offer.

## Methods

**Preparation of scattering layers.** SiO$_2$ particles of 0.2, 0.5, 1, 1.5, 3, and 4.5 wt.% (weight percentage of particles to host medium) were dispersed in a host material (NOA 73, Norland Products Inc.). SiO$_2$ particles with a diameter of 800 nm (SG-SO800) were supplied by Sukgyung AT Co., Ltd., Korea. Similarly, TiO$_2$ particles with a diameter of 200 nm (SG-TO200, Sukgyung AT Co., Ltd.) of 0.05, 0.1, 0.2, 0.5, 1, and 1.5 wt.% were dispersed in the same host material to realize scattering layers with a moderate degree of forward scattering. Each solution was homogenized using an ultrasonic liquid processor (VCX-750, Sonics & Materials, Inc.) at the power of 300 W for 15 min. Next, each solution was filtered by a PTFE syringe filter with 5 μm pores (Whatman™ 13 mm Puradisc Syringe Filters, 6784-1350, GE Healthcare) to remove particle aggregates that may have been present. The solutions were then placed under vacuum for 15 min to eliminate air bubbles. Each solution was bar-coated with the wet thickness of 300 μm defined by the application of an SUS plate. The films were UV cured for 5 min for complete solidification. The average thickness of the scattering layers after UV treatment was 287 (±11) μm (see Supplementary Fig. 12 for the scanning electron microscopy images).

**Measurement of the angular intensity of scattered light.** After at least 30 min of stabilization, normally incident light from a 532-nm-laser diode (DJ532-10, Thorlabs, Inc.) was applied to the scattering samples. Then, a photodiode on a goniometer was used to measure the angular intensity of transmitted scattered light passing through a scattering layer. The intensity was considered directly proportional to the current from the photodiode assuming that the spectrum was uniform over all angles. For other colors, 405-nm- and 635-nm-laser diodes (L405P20 and L635P5, Thorlabs, Inc.) were used (see Supplementary Fig. 11).

**Optical simulation.** Trans-scale optical simulation was realized by combining two custom-made MATLAB codes describing both RTT for scattering layers[20,22,26] and coherent dipole radiation theory also termed as power-dissipation model for OLED stacks[24,25,34]. The latter takes into account Purcell effect, dipole orientation, and coupling to waveguide and surface plasmon polariton modes and exhibits a quantitative match with experimental data in angle-resolved and spectrally resolved manner[16,21,34]. In the coherent optical simulation for an OLED stack, the emission zone was assumed to be located at the center of its emitting layer. Electrical loss was assumed to be absent[33].

The core part of Mie theory simulation code used to find conditions for high-$g$ scattering was taken from Mätzler and modified to reflect the needs of this work[36]. Scattering layers were assumed to be free from photon absorption throughout this study. In the simulation based on RTT, Henyey−Greenstein phase function was employed in lieu of full Mie theory-based phase function for simple and fast calculation[21,26]. Boundary conditions for the interface between a scattering layer and air were specified using Fresnel equations. Since RTT relies on the statistical nature of bulk-scattering, propagation within scattering layers were assumed incoherent. To confirm the validity of the custom-made RTT simulation code, the extraction efficiency and normalized angular outcoupled intensity of the RTT simulation was compared with those obtained for Monte-Carlo/ray-tracing simulation done with commercial software LightTools$^{TM}$. As shown in Supplementary Fig. 13, the results from the two methods agreed well, which demonstrates the accuracy and reliability of the realized RTT simulation. All simulations were done in the spectral range from 450 to 800 nm with a 1 nm interval, unless specified otherwise.

**OLED fabrication and characterization.** OLED devices were fabricated on 25 mm × 25 mm glass substrates with half-patterned 150-nm-thick ITO layers (AMG, Korea) to study EQE over various values of $S$ (Type A). For fabrication of devices utilizing the scattering-based light extraction to its full potential, 40 mm × 70 mm glass substrates were used instead to accommodate a large back reflector and to avoid interference from adjacent devices or electrodes for contacts (Type B). The substrates were cleaned with deionized water, deionized water with detergent, isopropyl alcohol, and acetone for 15 min, respectively, in an ultrasonic bath. Then, the substrates were treated with UV-plasma for 1 min. Immediately after UV treatment, the substrates were transported to a vacuum thermal evaporator, and all the layers in the OLEDs were deposited under a pressure less than 5×10$^{-6}$ torr without breaking the vacuum. OLEDs under study had the following structure: ITO (150 nm)/TAPC (130 nm for Ir(dmppy-ph)$_2$tmd; 90 and 80 nm for Ir (ppy)$_2$acac and Ir(ppy)$_3$, respectively)/TCTA (10 nm)/TCTA: B3PYMPM: Ir (dmppy-ph)$_2$tmd (4 wt.%, 30 nm) for orange OLEDs; Ir(ppy)$_2$acac (8 wt.%, 30 nm) or Ir(ppy)$_3$ (8 wt.%, 30 nm) for green OLEDs /B3PYMPM (65 nm)/LiF (1 nm)/Al (100 nm). Here, TAPC, TCTA, Ir(dmppy-ph)$_2$tmd, Ir(ppy)$_2$acac, Ir(ppy)$_3$, and B3PyMPM refers to di-[4-($N,N$-di-p-tolyl-amino)-phenyl]cyclohexane, 4,4′,4″-tris (carbazol-9-yl)triphenylamine, bis(2-(3,5-dimethylphenyl)-4-phenylpyridine) Ir (III) (2,2,6,6-tetramethylheptane-3,5-diketonate), bis(2-phenylpyridine)

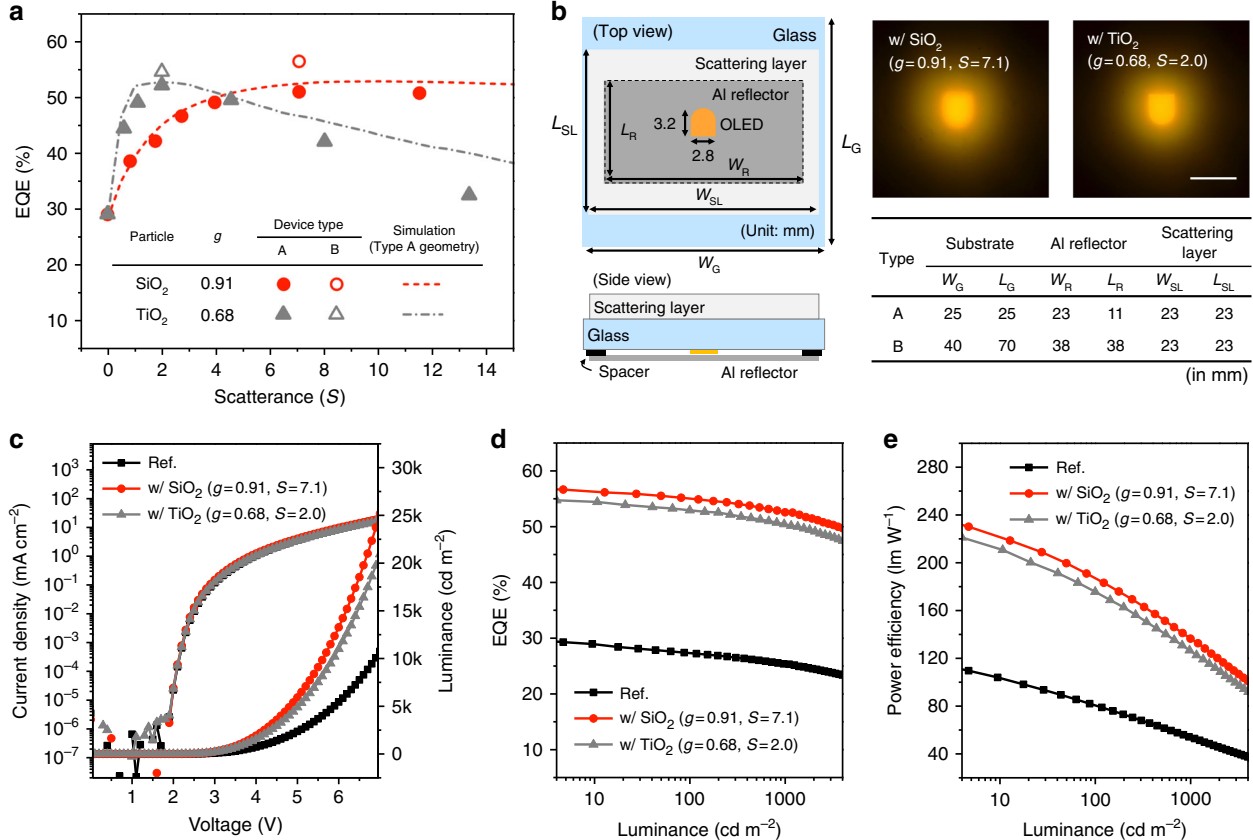

**Fig. 4** The performance of organic light-emitting diodes (OLEDs) fabricated on large substrates. **a** The measured external quantum efficiencies (EQEs) vs. scatterance ($S$) of OLEDs with SiO$_2$-based (asymmetry parameter ($g$) = 0.91) and TiO$_2$-based ($g$ = 0.68) scattering layers. The color-matched dotted lines indicate LightTools simulation results that reflect the actual geometry (Type A) used for EQE-$S$ trend study. **b** Device test structures used to mimic a large-area device with a small, lab-scale active area. Type A uses a small substrate so that one can make several samples at the same time for studying the overall trend over certain variables. Type B uses a large substrate to explore the ultimate light extraction achievable via external scattering. Photographs of OLEDs (Type B) are also shown for both SiO$_2$-based ($S$ = 7.1) and TiO$_2$-based ($S$ = 2.0) OLEDs (scale bar: 5 mm). **c–e** Device characteristics of Type B champion OLEDs

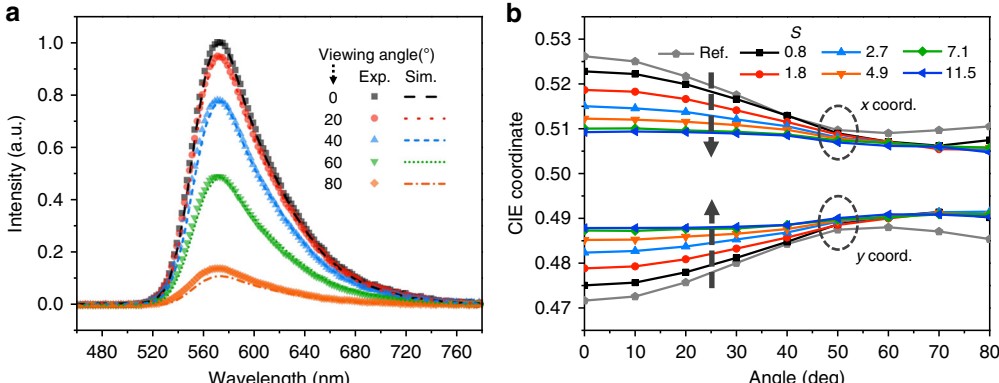

**Fig. 5** Angular spectral dependence of organic light-emitting diodes (OLEDs) with the SiO$_2$ scattering layers. **a** Experimental (semitransparent symbols) and simulated (dashed lines) normalized angular spectra of OLEDs with a 3 wt.% scattering layer (scatterance ($S$) = 7.1). **b** CIE coordinates of angular spectra of the devices with different SiO$_2$-based scattering layers

(acetylacetonate) iridium(III), tris(2-phenylpyridine)iridium(III), and 4,6-bis(3,5-di(pyridin-3-yl)phenyl)-2-methylpyrimidine, respectively. All organic materials except Ir(dmppy-ph)₂tmd were purchased from Nichem Fine Technology co. Ltd. The phosphorescent emitter, Ir(dmppy-ph)₂tmd, was provided from Y.-H.K. and K.-H.K. Area of all devices were approximately 8.0 mm² and were estimated carefully from dimensions measured with a digitized vernier calipers.

The scattering layers were optically coupled to the glass substrates at the opposite side of OLED devices using index matching fluid having a refractive index of 1.52 (IML 150, Norland Products Inc.). The current density, luminance, electroluminance (EL) spectra were measured using a programmable source meter (Keithley 2400), a calibrated Si photodiode, and a fiber optic spectrometer (EPP2000, StellarNet Inc.) in a nitrogen-filled glove box. The photodiode and spectrometer were mounted on a custom-made goniometer to measure the angular EL intensity and spectra. The EQE, power efficiency, and current efficiency were calculated by fully accounting for the measured angular distribution of the EL intensity and spectra. When a scattering layer was applied to an OLED, a 100-nm-thick aluminum back reflector (23 mm × 11 mm for Type A or 38 mm × 38 mm for Type B) prepared on an Si wafer for Type A case and 40 mm × 40 mm glass for type B case was placed underneath the OLED to minimize the amount of light that leaks through the area without a cathode due to the finite dimension of the device area (see discussion in the main text and Fig. 4b). This reflector was essential in reducing the gap between the experiment and simulation, the latter of which assumed an infinitely large device area.

**Data availability**. The authors declare that the data supporting the findings of this study are available within the article and its Supplementary Information files. Numerical values of data shown as graphs are available from the corresponding authors upon reasonable request.

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

## Acknowledgements

This work was supported in part by the National Research Foundation (NRF) of Korea funded by the Ministry of Science and ICT (NRF-2017R1A2B3010484 and NRF-2016M3A7B4910631); and in part by a research program "Technology Development of Low Cost Flexible Lighting Surface", which is a part of the R&D program of Electronics and Telecommunications Research Institute (ETRI). We are grateful to Sukgyung AT Co., Ltd., Korea for providing SiO₂ and TiO₂ nanoparticles.

## Author contributions

S.Y. and J.S. conceived an idea on trans-scale optical simulation combining the coherent dipole radiation theory and the RTT for global, multivariable high-speed modeling. J.S. made custom MATLAB codes for such trans-scale simulation and carried out all the simulations presented in this article. Based on simulation results, J.S. fabricated and analyzed scattering layers. E.K. assisted J.S. in performing LightTools™ simulations to ensure the validity of the proposed simulation. J.-J.K. and S.Y. made an idea of demonstrating highly efficient OLEDs using a scattering layer and a highly horizontally oriented emitter based on the result of trans-scale optical simulation. C.-K.M. discussed with J.S. in verifying a code based on the coherent dipole radiation model in OLEDs. Y.-H.K. and K.-H.K. provided the phosphorescent emitter. J.S. and K.-H.K. fabricated OLEDs and tested them with scattering layers. J.S. characterized the fabricated OLEDs. S.

Y. and J.S. analyzed all the data and wrote the manuscript and S.Y. coordinated all the experiments. All authors read and have given their approval to the final version of the manuscript.

## Additional information

**Competing interests:** The authors declare no competing interests.

