## [Peer Review File · Nature Communications]

Reviewers' comments:

Reviewer #1 (Remarks to the Author):

The paper is interesting and well written and should be accepted for publication following clarifications, addressing the following:

- The paper emphasizes the high EQE obtained with a scattering film however the >50% EQE was obtained for emitting dipoles that are highly horizontally oriented. As such the title is somewhat misleading; the >50% EQE is not an effect of the scattering film only. It would have been interesting to see the experimental results for other emitters, rather than the synthesized molecules studied.
- In relation to the above comment, the authors should compare their results to those described in previous works. For example, ACS photonics 2, 136 (2015). In the latter, a high refractive index flexible $\sim 2 \mu\text{m}$ host polymer with air voids as scattering centers was shown to enhance the EQE of green and white emitting OLEDs. The performance enhancement was attributed to the large difference between the refractive indices of the host and voids. In another example, TiO₂ nanoparticle-based scattering films under the anode (flexible substrate) were used for outcoupling enhancement in various structures (JAP 113, 204502 (2013)). In the current paper, the refractive index of the host layer used in the calculations was 1.57, and it is stated that the calculations revealed that the refractive index of the scattering particles should be similar to that of the host. What exactly does similar mean? How does this similarity of the refractive indices affect the effectiveness of the scattering process in the theory?
- The wet thickness of the scattering layer was 300 μm . This is a thick layer – what is the effect of the thickness of the scattering layer? Is it related to the similarity of the refractive indices of the particles and the host layer?

Reviewer #2 (Remarks to the Author):

In the present manuscript NCOMMS-17-33001-T, the authors investigate the combination of scattering layers and organic light-emitting diodes (OLEDs) for maximizing the external quantum efficiency of the latter. To do so, the authors implement a global optimization taking into account the scattering properties and the OLED thin film architecture in an holistic approach. Guided by optical simulations, the authors realize green OLEDs with slightly more than 50% EQE, which is a great achievement. In my view, the work shows excellent device performance and puts light onto a very much needed global efficiency optimization with focus on scalable out coupling structures. Only in discussion of the optimization, I see some points which could be improved during a revision, which I would suggest. Assuming these addressed, I see this manuscript on a clear track for suitability in Nature Communications. Please find below my comment.

- It is discussed that the anisotropy g is important for the overall effectiveness of the scattering layers used as outcoupling layers. In the Mie scattering framework, the particle dependence suggests that the larger the particles, the more forward directed the scattering is. Unfortunately, this work only uses 800 nm particles in their study, which fall into the plateau of high anisotropy as shown in Figure 2a. This work would benefit much from the inclusion of experimental results based on smaller diameter particles to validate the global optimization.

- Similarly to the above point, the results were obtained using nanoparticles with $n=1.46$, while the embedding host material has a refractive index of $n=1.57$, which overall seems to be a comparably small refractive index contrast. Is this the optimum? Is the overall performance also depending on the relation of the two materials refractive index? This discussion should be part of this work.

- The statement in line 76: „In the proposed device configuration, a EQE reached 56.2% ...„ it is

not clear to the reader, if this value is from simulation of experimental result. Looking at Figure 1, this finally clarifies as an simulation result, but it should be clear in the text as well.

- In the device description (methods) of the OLED stack, there is a typo: TCTA without space.
- The variation of the ITO thickness is discussed in the manuscript in detail and takes much of the visual attention in Figure 1, but even the authors conclude that the variation is very minor. I would suggest to shift this discussion (dependence on d_{ITO}) to the Supplementary Information, as it is not of high importance here.
- The complete manuscript is based on the optimization of green OLEDs. The characterization of the scattering layer is fittingly analyzed using a green 532 nm laser. Still, it would be much more compelling, if the authors would be able to discuss possible effects of dispersion (or scattering dependence on the wavelength in the visible range (blue to red color)).
- Looking at the EQE values of table 1 where the values are given at 10 and 1000 cd/m², which is greatly appreciated, a clear trend from reference to higher S values with respect to the EQE roll-off is observed. If one devices EQE₁₀₀₀/EQE₁₀, the values monotonically increase from 0.90 to 0.94 with increasing S, i.e. the roll-off in the devices becomes smaller. It would be nice, if the authors could check, if this really is a trend, and if so, why?
- The authors state the wet layer thickness of the as-prepared scattering layers. However, it would be nice to know, what the final thickness is.

Reviewer #3 (Remarks to the Author):

This manuscript is not recommended for publication, as the originality/scientific merit to be suitable for publication in Nature Communications is insufficient. Similar trans-scale optical simulations using Henyey-Greenstein phase functions was previously reported by the same group (DOI: 10.1002/adma.201404862). Also, external light scattering medium and horizontally oriented dipole emitters are not new scientific findings in the field.

One interesting point to note is the author's finding based on the use of silicon dioxide nanoparticles ($n=1.46$), which has a similar refractive index as its host media, NOA73 ($n=1.57$). This is counter-intuitive to typical light scattering films, and the authors explained this choice to achieve a high g (asymmetry parameter). The accuracy of this simulated result is difficult to argue against based on the limited information in the article. An in-depth explanation on the calculation of the g parameter and device performance comparisons against higher n nanoparticles in the same OLED structure would provide much stronger support for the author's argument.

The device characteristics graphs also raise a few questionable behaviors. It is interesting that the highest EQE in Figure 3 and Table 1 is $S = 7.1$ at 50.9% while $S = 11.5$ shows higher PE and CE than $S = 7.1$. In addition, the inconsistent current density in Figure 3a (below the turn on voltage at less than 2V), is also puzzling.

Response to Reviewers' Comments and Summary of the Changes

We deeply thank the reviewers for their thoughtful comments and suggestions for improvements. Responses to each of the comments are summarized below. As the changes in the manuscript are rather massive, let us provide **a table summarizing the revised parts at the end**, instead of showing all the revised sentences. The red-lined version of the revised manuscript will also be provided so that reviewers can easily recognize the revised parts.

Reviewer #1's comments:

The paper is interesting and well written and should be accepted for publication following clarifications, addressing the following:

Comment #1

The paper emphasizes the high EQE obtained with a scattering film; however, the >50% EQE was obtained for emitting dipoles that are highly horizontally oriented. As such the title is somewhat misleading; the >50% EQE is not an effect of the scattering film only. It would have been interesting to see the experimental results for other emitters, rather than the synthesized molecules studied.

Authors' response to Reviewer #1's Comment #1

We are grateful to Reviewer#1 for his or her considerate advice. As suggested, we first modified the title of the manuscript to "Lens-free OLEDs with over 50% external quantum efficiency via tailored external scattering and horizontally oriented dipole emitters" and applied the proposed method to OLEDs with Ir(ppy)₂acac and Ir(ppy)₃ - widely studied phosphorescent emitters that are readily available from commercial vendors. Note that Ir(ppy)₂acac has horizontal dipole ratio of 0.76 (i.e. slight preference toward horizontal orientation) and that Ir(ppy)₃ has horizontal dipole ratio of 0.67 (i.e. isotropic arrangement)^{R1}.

Global optimization of their device structures was also done with the proposed external scattering. In the experiments done with the optimized device structures, we were able to realize EQE as high as 44.8% and 40.3% for Ir(ppy)₂acac-based and Ir(ppy)₃-based OLEDs, respectively. (**Fig. R1** and **R2**.) These values are among the highest ever achieved for these emitters with external scattering approach. We believe this addition will let readers easily make a comparison with many of the already published works obtained with these well-known emitters and better grasp how much contribution comes from scattering and how much from horizontal orientation of emitters.

Figure R1 | Summary of studies using two kinds of well-known emitters with a different degree of horizontal dipole orientation: Ir(ppy)₂acac ($\Theta=0.76$) and Ir(ppy)₃ ($\Theta=0.67$). (a, c) Global optimization done for OLEDs based on (a) Ir(ppy)₂acac and (c) Ir(ppy)₃, respectively. The RTT simulation shows that $\eta_{EQE}^{(max)}$ can reach 49.2% when $d_{ETL}=65$ nm, $d_{HTL}=90$ nm for Ir(ppy)₂acac and 43.8% when $d_{ETL}=65$ nm, $d_{HTL}=80$ nm for Ir(ppy)₃ with the optimal d_{ITO} , g , and S . (b, d) Measured EQE of OLEDs with the proposed SiO₂-based scattering layers as a function of luminance. (b) Experimental EQE of Ir(ppy)₂acac-based OLEDs. (d) Experimental EQE of Ir(ppy)₃-based OLEDs.

Figure R2 | Measured EQEs vs. to S of the SiO₂-based scattering layer and an emitter of OLEDs.

Comment #2

In relation to the above comment, the authors should compare their results to those described in previous works. For example, ACS photonics 2, 1366 (2015). In the latter, a high refractive index flexible $\sim 2 \mu\text{m}$ host polymer with air voids as scattering centers was shown to enhance the EQE of green and white emitting OLEDs. The performance enhancement was attributed to the large difference between the refractive indices of the host and voids. In another example, TiO₂ nanoparticle-based scattering films under the anode (flexible substrate) were used for outcoupling enhancement in various structures (JAP 113, 204502 (2013)). In the current paper, the refractive index of the host layer used in the calculations was 1.57, and it is stated that the calculations revealed that the refractive index of the scattering particles should be similar to that of the host. What exactly does similar mean? How does this similarity of the refractive indices affect the effectiveness of the scattering process in the theory?

Authors' response to Reviewer #1's Comment #2

(1) Overview – Influence of asymmetry parameter (g) on the trend of the maximum EQE vs. scatterance (S).

We thank Reviewer#1 for this insightful comment. When one tries to utilize a scattering phenomenon for outcoupling enhancement, it is quite natural to start with a combination that can lead to a sufficient degree of scattering effect. As scattering occurs due to geometrical objects (e.g. particles, voids) that have an optical property different from that of a surrounding medium, it is reasonable to choose a geometrical object (hereafter 'scatterers') from those having high contrast in refractive index from the surrounding medium. It is a main reason why voids ($n=1$) or TiO₂ particles ($n=2.5$) were used as scatterers in the previous works mentioned by Reviewer#1 (ACS Photonics, 2015; J. Appl. Phys., 2013).

The high index-contrast achieved in those systems does allow each scattering event to change the direction of light propagation significantly, helping extract the light that would otherwise be confined inside substrate or ITO/organic layers. However, we should keep in mind that scattering events (particularly those yielding a large deflection angle) could, in some cases, become an obstacle to the light that would normally be outcoupled. Therefore, enhancing the degree of scattering can be justified only when it improves *the net outcoupling efficiency (η_{out}) with both positive and negative effects taken into account.*

As shown in **Fig. 1e** in the original manuscript (shown below as **Fig. R3a**), scattering films with high or low asymmetry parameter $g = \langle \cos \theta_{sc} \rangle$ impacts differently on the maximum EQE ($\eta_{EQE}^{(max)}$) as scatterance (S) increases:

- (i) those with g close to 0 ('near-isotropic scattering') induce a steep increase in η_{out} when S increases from 0 until a certain optimum S ($=S_{opt}$; around 1) is reached but tend to yield a rapid decrease in $\eta_{EQE}^{(max)}$ after S_{opt} . This is a clear indication that the increased back-scattered portion, which can have a negative effect on outcoupling, quickly outweighs the positive effect on outcoupling even in a low scatterer concentration when scattering layers with g close to 0 are used.
- (ii) those with g close to 1 ('forward-intensive scattering') show a relatively slow yet almost monotonous increase in $\eta_{EQE}^{(max)}$ until S hits 3-4, after which $\eta_{EQE}^{(max)}$ stays almost unchanged. Key consequences are that (a) devices with forward-intensive scattering films can achieve higher η_{EQE} upon global optimization including S than devices with near-isotropic scattering films; and that (b) η_{EQE} can stay almost flat near its global maximum over a significantly wide range of S , which is

regarded beneficial from the practical standpoint.

Figure R3 | Characteristics of scattering depending on asymmetry parameter. (a) Calculated maximum EQE ($=\eta_{\text{EQE}}^{(\text{max})}$) done for OLEDs under study vs. S for $g = 0, 0.5,$ and 0.95 under study in the manuscript. Here, the color-matched arrows indicate the range of S that can lead to $\eta_{\text{EQE}}^{(\text{max})} > 50\%$ for each of the cases under study. (b) Cartoons showing the angular distribution of a scattered light from a scatterer-host combination having g of 0 (isotropic) or a value in between 0 and 1 (forward scattering).

But what about a scattering film with moderate forward scattering, for example, those with g around 0.5? In such a case, $\eta_{\text{EQE}}^{(\text{max})}$ exhibits a tendency similar to those with g close to 0, but the global maximum for EQE is higher than that obtainable with $g = 0$ and, in fact, is fairly close to that achievable with the scattering films with g close to 1. Furthermore, $\eta_{\text{EQE}}^{(\text{max})}$ close to its global maximum can be achieved in a wider range of S than the film with g close to 0, although the corresponding range of S is still narrower than OLEDs adopting the scattering films with g close to 1 and $\eta_{\text{EQE}}^{(\text{max})}$ exhibits a rather steep variation around the optimal S ($= S_{\text{opt}}$).

In summary, films consisting of a host/scatterer combination with moderate-to-highly forward scattering will generally be advantageous in achieving the highest $\eta_{\text{EQE}}^{(\text{max})}$ that can be realized with external scattering films. In case of scattering films with moderate forward scattering, a special care must be paid so that S_{opt} is first identified for a given OLED structure, and then the concentration of scatterers or the thickness of the scattering layer made thereof is tightly controlled so that S may be close to S_{opt} .

(2) Regarding the “similarity” requirements for the refractive indices of scatterers (n_{SP}) and a host medium (n_{SL}).

Once we accept the importance of having high g values in maximizing the outcoupling efficiency, it becomes natural to ask the following question: *how similar should n_{SP} and n_{SL} be?* At first glance at **Fig. 1e**, it seems the maximum EQE continues to increase as g approaches unity; however, one should keep in mind that a certain set of conditions will be meaningful only if it can be met with physical parameters that are within practically reasonable range.

To respond to the question regarding this requirement for “similarity” between n_{SP} and n_{SL} , we have further carried out Mie calculation for the following parameters as a function of both n_{SP} and diameter (d_{SP}) with n_{SL} fixed as 1.57 (**Fig.R4**):

- asymmetry parameter (g)

- scattering efficiency (Q_{sc}), defined as scattering cross section (σ_{sc}) divided by geometrical cross section ($\pi d_{SP}^2/4$)
- scattering cross section (σ_{sc})
- maximum scattering coefficient ($\mu_{sc}^{(c,max)}$) for cubic arrangement, defined by:

$$\mu_{sc}^{(c,max)} = 1/L_{MFP}^{(c,min)} = N_{SP}^{(c,max)} \sigma_{sc} = \sigma_{sc}/d_{SP}^3 = Q_{sc} (\pi d_{SP}^2/4) / d_{SP}^3 = (\pi/4) Q_{sc} / d_{SP}$$

where $N_{SP}^{(c,max)}$ is the maximum number density attainable for simple cubic arrangement of spherical particles (i.e. one particle per cubic unit cell with a side length of d_{SP}) with the diameter of d_{SP} and $L_{MFP}^{(c,min)}$ is the minimum mean-free path corresponding to such a case.

Figure R4 | Various Mie scattering parameters obtained as a function of d_{SP} and n_{SP} for host medium with $n_{SL} = 1.57$ at the wavelength (λ) of 532 nm. (a-d) (a) asymmetry parameter (g) (b) scattering efficiency (Q_{sc}) (c) scattering cross section (σ_{sc}) (d) maximum scattering coefficient ($\mu_{sc}^{(c,max)}$) for cubic arrangement.

The following important observations can be made from the results shown in **Fig. R4**:

- To obtain moderate forward scattering ($g \sim 0.5$), d_{SP} should be no smaller than ca. 150 - 200 nm.
- To obtain highly forward scattering (e.g. $g > 0.8-0.9$), d_{SP} should be large and $|n_{SP}-n_{SL}|$ should be not too large. For example, g can be larger than 0.9 when $d_{SP} >$ ca. 500 nm and $|n_{SP}-n_{SL}| <$ ca. 0.25).
- In consideration of the scattering efficiency (Q_{sc}), $|n_{SP}-n_{SL}|$ may not be made arbitrarily small. The minimum difference in $|n_{SP}-n_{SL}|$ has to be set by looking at the constraints on the thickness or concentration of a scattering layer.

For example, when $n_{SP}=1.56$ ($|n_{SP}-n_{SL}| = 0.01$) for a host medium with $n_{SL} = 1.57$ and a particle with $d_{SP} = 1 \mu\text{m}$, Q_{sc} becomes as low as 6.7×10^{-3} ; in this case, $\mu_{sc}^{(c,max)}$ and the corresponding $L_{MFP}^{(c,min)}$ becomes $5.2 \times 10^{-3} \mu\text{m}^{-1}$ and $191 \mu\text{m}$, respectively. Because $S = \ln(1/T_{inline}) = d_{SL}/L_{MFP}$, this means that at least $191 \mu\text{m}$ -thick scattering layer is required to achieve S of 1. Therefore, achieving S of 5-10 in such a case will require d_{SL} at least on the order of 1-2 mm. These values are for closely packed cases. For most practical range of concentrations, the required thickness will be much larger. Therefore, the actual required thickness is likely to be impractical in most of the cases.

In the present work using SiO_2 nanoparticles, $(d_{SP}, |n_{SP}-n_{SL}|) = (800 \text{ nm}, 0.11)$, which yields g close to 1 and, at the same time, a sufficient level of Q_{sc} ($= 0.45$) and $\mu_{sc}^{(c,max)} = 1/L_{MFP}^{(c,min)}$ ($= 0.44 \mu\text{m}^{-1} = 1/(2.3 \mu\text{m})$). These values mean that the minimum d_{SL} for $S = 5-10$ can be around 10 to a few tens of μm . Even with the ten-time lower concentration than the closely packed case, one could obtain $S = 5-10$ with the thickness of a scattering layer under a few hundreds μm , as demonstrated in the present work.

(3) Comparison with the previous works

It might be difficult to make a direct comparison with the previous works because the previous works are often not aimed at the full optical optimization and/or because base OLED structures are not necessarily the same. Nevertheless, we do believe that it will be of interest and great help to the field to compare them in the same condition and have them subject to the full global optimization proposed in this work. To this end, we have prepared scattering layers with various concentrations of TiO_2 nanoparticles (SG-TO200, Sukgyung AT Co., Ltd., Korea; $d_{SP} = 200 \text{ nm}$), which are readily available from commercial sources.

With these TiO_2 embedded layers, scattering layers with g of 0.68 were realized with various level of S , as can be seen in **Fig. R5**. The g value of 0.68 ± 0.03 is close to that predicted by Mie calculation ($g=0.60$) shown in **Fig. R4** and represents scattering layers with moderate forward scattering. When coupled with the optimal device structure based on the same emitter ($\text{Ir}(\text{dmpy-ph})_2\text{tmd}$), EQE of the devices with these scattering layers indeed exhibited the trend expected for moderate level of g ; that is, EQE increased with S until it reached the peak at S near 2.5 and decreased significantly as S further increased.

Figure R5 | Measured (dot) angular characteristic of transmitted scattered light passing through scattering layers using TiO_2 nanoparticles whose diameter (d_{SP}) is about 200 nm. The experimental data were fitted (dashed line) to

find asymmetry parameter (g). The value of g giving the nearest fitting result was 0.68. Inset: A SEM image of a fabricated TiO₂-based scattering layer. (Scale bar: 1 μm)

It is interesting to see that the maximum EQE of OLEDs with this TiO₂-based system was even slightly higher than the maximum EQE of OLEDs with scattering layers having g of 0.91. This seemingly contradicting result can be understood by noting that light extraction with scattering layers requires a relatively large area to utilize their full potential. Such a tendency is especially true when high- g scattering is used because substrate-confined light, travelling almost parallel to the substrate, will require a larger number of scattering events (i.e. longer travel) before it gets total effective deflection sufficient for outcoupling in such a system. (See **Fig. R6** for a schematic overview.)

Figure R6 | The schematic illustration of the effect of asymmetry parameter on the lateral traveling distance (d_{tr}) in parallel direction before outcoupling of light that is emitted into the scattering layer at high angle. (a) Case of isotropic scatterers ($g = 0$) (b) Case for the high degree of forward scattering ($g = \sim 0.9$).

In order to avoid this area-dependent issue, we have enlarged a substrate and a back reflector to $40 \times 70 \text{ mm}^2$ and $38 \times 38 \text{ mm}^2$, respectively, and modified a device layout so that electrodes for contacts, etc. are placed far away from the device area to avoid any optical blocking of the light being outcoupled. Results from this experiment shown in **Fig. R7** indicate that OLEDs with SiO₂ particles ($g=0.91$, $S=7.1$) exhibit EQE as high as 56.3% (red hollow rectangle) while those with TiO₂ particles ($g=0.68$, $S=2.0$) show EQE of 54.5% (red hollow triangle) at their respective optimum S ; both results are close to the max EQE values predicted for each case by the RTT-based trans-scale simulation. From the photographs, the lateral propagation distance is indeed larger for the OLED with the SiO₂-based scattering layer than for the OLED with the TiO₂-based scattering layer. (See **Fig. R8**)

If voids were used for scatterers with the present host medium, g and scattering coefficient could be in between those of TiO₂ systems and those of SiO₂ systems. Therefore, the maximum EQE achievable will also be likely to be in between the EQEs of those two systems. In summary, scattering layers based on SiO₂, TiO₂, or air voids are all capable of yielding very high EQE values, although there are small differences. However, it should be emphasized that these very high EQEs will be achievable only if their scatterance (S) values are carefully adjusted so that they fall on their respective optimal ranges that consider the optical characteristics of both the scattering layers and base OLED structures as a whole. It will be particularly important for systems with low-to-mid g scatterers as their optimal ranges for S are rather limited.

Figure R7 | The performance of OLEDs with the proposed scattering layers including devices fabricated on large substrates. (a) The measured EQEs vs. S of OLEDs with SiO₂-based ($g=0.91$) and TiO₂-based ($g=0.68$) scattering layers. The color-matched dotted lines indicate LightTools simulation results that reflect the actual geometry (Type A) used for EQE- S trend study. Inset: Photographs of SiO₂-based OLEDs according to S . **(b)** Device test structures used to mimic a large-area device with a small, lab-scale active area. Type A uses a small substrate so that one can make several samples at the same time for studying the overall trend over certain variables. Type B uses a large substrate to explore the ultimate light extraction achievable via external scattering. Photographs of OLEDs (Type B) are also shown for both SiO₂-based ($S=7.1$) and TiO₂-based OLEDs ($S=2.0$). **(c-e)** Device characteristics of Type B champion OLEDs.

Figure R8 | The enlarged version of the photographs shown in Fig. R7b.

Comment #3

The wet thickness of the scattering layer was 300 μm . This is a thick layer – what is the effect of the thickness of the scattering layer? Is it related to the similarity of the refractive indices of the particles and the host layer?

Authors' response to Reviewer #1's Comment #3

A scattering layer having high g requires high scatterance (S) to ensure the maximal outcoupling as shown in **Fig. 1e** in the main text. High S can be achieved either by increasing the number density of scatterers (ρ_{SP}) or by increasing the thickness of the scattering medium (d_{SL}). Of course, the processability may limit ρ_{SP} because too high density can result in aggregation. In such a case, the thickness of the host medium should be increased for a given ρ_{SP} . In the present work, ca. 300 μm -thick layer was used because, with this, we were able to achieve homogeneous scattering media with S of up to ca. 12; this was essential in gaining a full picture on the trend of EQE over a wide range of S . In practice, S of ca. 6 would already be high enough to get the highest EQE as shown in Fig. 1e; thus, the actual thickness of the scattering layer can be reduced by half if one uses the same concentration as the scattering layer with S of 12.

In an application where the thinness of a scattering layer is a critical constraint (e.g. ultra-flexible OLEDs), TiO_2 systems could be a better option because it can have a shorter L_{MFP} and thus thinner scattering layer for a target S ($\because d_{\text{SL}} = S \times L_{\text{MFP}}$).

Nevertheless, the system proposed in this work should still be suited for a wide range of applications based on glass-based OLEDs. Furthermore, the host medium itself can be used as a substrate; in this case, it will also be applicable to flexible devices as its thickness falls on the thickness range of typical plastic substrates. In any situations mentioned above, the global optimization approach presented in this work will provide a quantitative strategy that can balance the high-efficiency demand and the form-factor constraints.

Reviewer #2's comments:

In the present manuscript NCOMMS-17-33001-T, the authors investigate the combination of scattering layers and organic light-emitting diodes (OLEDs) for maximizing the external quantum efficiency of the latter. To do so, the authors implement a global optimization taking into account the scattering properties and the OLED thin film architecture in an holistic approach. Guided by optical simulations, the authors realize green OLEDs with slightly more than 50% EQE, which is a great achievement. In my view, the work shows excellent device performance and puts light onto a very much needed global efficiency optimization with focus on scalable out coupling structures. Only in discussion of the optimization, I see some points which could be improved during a revision, which I would suggest. Assuming these addressed, I see this manuscript on a clear track for suitability in Nature Communications. Please find below my comment.

Comment #1 & #2

- It is discussed that the anisotropy g is important for the overall effectiveness of the scattering layers used as outcoupling layers. In the Mie scattering framework, the particle dependence suggests that the larger the particles, the more forward directed the scattering is. Unfortunately, this work only uses 800 nm particles in their study, which fall into the plateau of high anisotropy as shown in Figure 2a. This work would benefit much from the inclusion of experimental results based on smaller diameter particles to validate the global optimization.

- Similarly to the above point, the results were obtained using nanoparticles with $n=1.46$, while the embedding host material has a refractive index of $n=1.57$, which overall seems to be a comparably small refractive index contrast. Is this the optimum? Is the overall performance also depending on the relation of the two materials refractive index? This discussion should be part of this work.

Authors' response to Reviewer #2's Comment #1 & #2

Authors are grateful to Reviewer#2 for his or her insightful comments and constructive suggestions. In fact, Reviewer#2's 1st and 2nd comments are in line with Reviewer#1's 2nd comment. Please refer to the discussion we made in response to Reviewer#1's Comment 2. There are (i) additional discussion based on simulation of not only g but also scattering efficiency (Q_{sc}) obtained as a function of both refractive index of a scattering particle (n_{SP}) and its diameter (d_{SP}); and (ii) additional experimental results based on TiO₂ nanoparticles having n_{SP} of ca. 2.5 and d_{SP} of ca. 200 nm.

Although we could not go through a full experimental sweep of n_{SP} and d_{SP} due to the limited availability of nanoparticles and time constraints, we hope description based on generalized parameters g and S could suffice for what Reviewer#2 intended us to cover. Experimental demonstrations done with scattering layers having high- and mid- g over their full ranges of S will also serve as a good illustration that emphasizes the importance of the global optimization made in this work.

Comment #3

The statement in line 76: „In the proposed device configuration, a EQE reached 56.2% ...”, it is not clear to the reader, if this value is from simulation of experimental result. Looking at Figure 1, this finally clarifies as an simulation result, but it should be clear in the text as well.

Authors' response to Reviewer #2's Comment #3

Authors agree with Reviewer#2 that the original expression could indeed be misleading. Hence, we revised the sentence to clearly indicate that the value (56.2%) refers to the *theoretical* maximum predicted by our simulation.

Comment #4

In the device description (methods) of the OLED stack, there is a typo: TCTA without space.

Authors' response to Reviewer #2's Comment #4

Authors are very much grateful to Reviewer#2 for his or her attention to great details. We have corrected our typo.

Comment #5

The variation of the ITO thickness is discussed in the manuscript in detail and takes much of the visual attention in Figure 1, but even the authors conclude that the variation is very minor. I would suggest to shift this discussion (dependence on d_{ITO}) to the Supplementary Information, as it is not of high importance here.

Authors' response to Reviewer #2's Comment #5

Authors are grateful to Reviewer#2 for his or her keen observation and constructive suggestion. It turned out that the reason that the effect of ITO thickness was minor was because the thicknesses of the other organic layers were also varied to compensate any negative effect induced by the varied ITO thickness. We have thus prepared a stacked plot consisting of a multiple of contour plots that were obtained with the *total* thickness of ITO plus organic layers* ($=d_{org}+d_{ITO}=d_{active}$) varied. The detailed device structure used in the study has also been moved from Supplementary Information for better understanding. For a given set of $\{d_{active}, g, S\}$, each data point in Fig. 1b of the main text was obtained by identifying a combination of d_{ITO} and the thicknesses of electron and hole transport layers (d_{ETL}, d_{HTL}) leading to the maximum achievable EQE ($=\eta_{EQE}^{(max)}$), for a given set of $\{d_{active}, g, S\}$. In the simulation, d_{ITO} was chosen to be no smaller than 50 nm to ensure the sheet resistance of ITO is not too large. The plot now indicates that there is an optimal d_{active} that is better suited for the maximal light extraction when coupled with scattering layers, as shown in **Fig. R9**. Power dissipation spectra

obtained for the base reference device structure indicate that the optimal case ($d_{\text{active}} = 385$ nm) corresponds to the case where the sum of air modes and substrate modes is at its maximum, rather than the case where only the air mode is maximized. (**Supplementary Fig. 3** in SI.). In the revised manuscript, we have thus replaced Fig. 1b with a new plot and added a relevant description in the main text and Supplementary Information.

Fig. R9 | The stacked contour plots of maximum achievable EQE ($\eta_{\text{EQE}}^{(\text{max})}$) vs. g and S obtained for various d_{active} values. In each case, d_{ITO} was made not smaller than 50 nm. Each data points were obtained by finding a combination of d_{ITO} and the thicknesses of electron and hole transport layers (d_{ETL} , d_{HTL}) leading to the maximum achievable EQE ($=\eta_{\text{EQE}}^{(\text{max})}$) for a given set of $\{d_{\text{active}}, g, S\}$. The global maximum considering all the parameters is presented by the black star.

Comment #6

- The complete manuscript is based on the optimization of green OLEDs. The characterization of the scattering layer is fittingly analyzed using a green 532 nm laser. Still, it would be much more compelling, if the authors would be able to discuss possible effects of dispersion (or scattering dependence on the wavelength in the visible range (blue to red color)).

Authors' response to Reviewer #2's Comment #6

Unlike Rayleigh scattering, whose scattering cross section depends sensitively on the wavelength, Mie scattering is known to be relatively color neutral, which can also be confirmed in **Fig. R10** shown below. Such characteristics will hold well even if one takes dispersion of optical constants into a full account, because many of the glassy materials tend to exhibit a similar trend in their dispersion relations; i.e. individual n_{SP} and n_{SL} do vary with λ , but both of them tend to increase as λ gets smaller, making $(n_{\text{host}} - n_{\text{N}})$ remain almost unchanged in many cases. On the other hand, the ratio of d_{SP} to λ becomes smaller as λ increases, and thus g tends to decrease with λ . Nevertheless, the change remains minor in the visible spectral range. We have added a short paragraph discussing the effect of dispersion to the main text along with relevant graphs added in Supplementary Information.

Figure R10 | Calculated (Cal.) and experimentally obtained (Exp.) asymmetry parameter of scattering layers as a function of wavelength. The calculation was performed based on Mie theory.

Comment #7

Looking at the EQE values of table 1 where the values are given at 10 and 1000 cd/m², which is greatly appreciated, a clear trend from reference to higher S values with respect to the EQE roll-off is observed. If one devices EQE₁₀₀₀/EQE₁₀, the values monotonically increase from 0.90 to 0.94 with increasing S , i.e. the roll-off in the devices becomes smaller. It would be nice, if the authors could check, if this really is a trend, and if so, why?

Authors' response to Reviewer #2's Comment #7

Authors thank Reviewer#2 for pointing out the important characteristics authors initially had missed. As OLED devices under comparison were essentially the same, such a differing trend in the roll-off characteristics indeed sounded unreasonable at first sight. On second thought, however, we realized that it was mainly because comparison was being made *at a specified luminance*, not at a specified current density. Note that higher S leads to enhanced outcoupling in the proposed scattering layers with the relatively high g . That is, the higher S is, the lower is the required current density for a given luminance in the proposed device scheme. Therefore, the roll-off characteristics should be less severe in the OLEDs with high S scattering layers in this work if comparison is made at specified luminance levels. Such a trend, however, may not necessarily be true if comparison is made at specified current densities.

To verify this notion, EQEs of the devices under study are summarized at several *current densities* in Table R1. The table shows that there is no significant correlation between roll-off characteristic and S of the scattering layers when comparison is made at specified current densities (J).

Table R1 | External quantum efficiency of the OLEDs at different current density values.

S	External quantum efficiency (%)		
	At $J = 0.1$	1	10
Ref.	27.9 (1)*	26.3 (0.943)	23.4 (0.839)
0.8	37.1 (1)	35.1 (0.946)	31.2 (0.841)
1.8	40.7 (1)	38.6 (0.948)	34.4 (0.845)
2.7	45 (1)	43 (0.956)	38 (0.844)
4.9	47.7 (1)	45.5 (0.954)	40.4 (0.847)
7.1	49.9 (1)	47.6 (0.954)	42.1 (0.844)
11.5	49.7 (1)	47.3 (0.952)	41.6 (0.837)

*: values in parentheses denote relative value compared to the value at $J = 0.1 \text{ mA cm}^{-2}$

From the practical perspectives, the results in Table 1 well illustrate the importance of having an efficient outcoupling structure, as a target brightness level can be achieved at a relatively low current density, which would be less subject to a severe roll-off.

Comment #8

The authors state the wet layer thickness of the as-prepared scattering layers. However, it would be nice to know, what the final thickness is.

Authors' response to Reviewer #2's Comment #8

The final thickness of the scattering layers fabricated in this work was measured with the scanning electron microscopy (SEM), as shown in **Fig. R11**. The average thickness measured from 12 positions with four samples turned out to be $287 (\pm 11) \mu\text{m}$. The relatively small difference from the wet thickness is considered to result from the “fast cure” characteristics of Norland optical adhesive (NOA) used in this work; its specification sheet indicates that it sets in seconds upon UV exposure during the initial soft curing. In this way, its initial wet thickness can be well maintained even though UV exposure continues for full curing.

Figure R11 | The scanning electron microscopy (SEM) images of the cross section of fabricated scattering layers.

Reviewer #3's comments:

Comment #1

This manuscript is not recommended for publication, as the originality/scientific merit to be suitable for publication in Nature Communications is insufficient. Similar trans-scale optical simulations using Henyey-Greenstein phase functions was previously reported by the same group (DOI: 10.1002/adma.201404862). Also, external light scattering medium and horizontally oriented dipole emitters are not new scientific findings in the field.

Authors' response to Reviewer #3's Comment #1

We fully understand Reviewer #3's concern, and we think others may get a similar impression when they first see our work. This isn't surprising at all because many groups (including us) have already reported on scattering-based light extraction schemes in OLEDs. However, we should keep in mind that it is often worthwhile to re-visit a technology that 'we believe' we know, especially if it can identify the factors that have not been well recognized but can be quite critical in unlocking the *full potential* of the technology, all in a scalable and practically meaningful way.

The biggest difference of the present work from our own previous report in Advanced Materials (DOI: 10.1002/adma.201404862) is that optical modeling in the present work does not rely on Monte-Carlo simulation for the light propagation within a scattering layer but instead use a *full equation-based approach*. While the former is all right for analyzing a few selected systems and maybe for parametric studies with one or two parameters varied for a specific base structure, its time-consuming nature makes it less suitable for the *global* analysis where the multitude of different parameters (of both base device structure and scattering layers) are varied at the same time. The holistic and Monte-Carlo-free equation-based nature of the present work enables one to consider the effect of both an underlying thin-film structure and a scattering layer in a facile yet comprehensive fashion, revealing (i) how high EQE of OLEDs can ultimately be when one adjusts the numerous physical parameters of participating thin films or, in combination, those of scattering layers and (ii) what the most important parameters are in outcoupling standpoint.

With the revision made in response to Reviewer #1's Comment #2 (Reviewer #2's Comments #1 and #2; Reviewer #3's Comment #1), the depth and breadth of information provided by this work has become far more significant than that in our previous work. Looking at the *combined* effect of a scattering layer and the horizontal dipole ratio of emitters is in fact a good example showing the value of the present work. Because the horizontal dipole orientation has a relatively high outcoupling efficiency already from the beginning and because combination of two outcoupling methods often does not yield the enhancement factor given by simple multiplication of the individual enhancement factors, it is not trivial at all to predict how much enhancement can be ultimately achieved when horizontal dipole orientation is combined with external outcoupling structures.

Nevertheless, we agree with Reviewers that our original manuscript needed clarification and had a room for improvement. With revisions made according to all the reviewers' invaluable comments and suggestion, we hope the revised manuscript can be regarded eligible for publication in Nature Communications.

Comment #2

One interesting point to note is the author's finding based on the use of silicon dioxide nanoparticles ($n=1.46$), which has a similar refractive index as its host media, NOA73 ($n=1.57$). This is counter-intuitive to typical light scattering films, and the authors explained this choice to achieve a high g (asymmetry parameter). The accuracy of this simulated result is difficult to argue against based on the limited information in the article. An in-depth explanation on the calculation of the g parameter and device performance comparisons against higher n nanoparticles in the same OLED structure would provide much stronger support for the author's argument.

Authors' response to Reviewer #3's Comment #2

Authors are grateful to Reviewer#3 for his or her keen observation and insightful comments. We have carried out the requested in-depth analysis and experiments including (i) Mie calculation of various scattering parameters (g , scattering efficiency, etc.) obtained as a function of both refractive index of a scattering particle (n_{SP}) and its diameter (d_{SP}); and (ii) experimental results based on TiO_2 nanoparticles having refractive index of ca. 2.5 and diameter of about 200 nm. This comprehensive study led us to get a more balanced view on the optimization strategy for the scattering-based light extraction, which we believe will do the same to the readers in this field.

Reviewer#3's Comment#2 is in line with Reviewer#1's 2nd comment. Please refer to the discussion we made in response to Reviewer#1's Comment 2.

Comment #3

The device characteristics graphs also raise a few questionable behaviors. It is interesting that the highest EQE in Figure 3 and Table 1 is $S = 7.1$ at 50.9% while $S = 11.5$ shows higher PE and CE than $S = 7.1$. In addition, the inconsistent current density in Figure 3a (below the turn on voltage at less than 2V), is also puzzling.

Authors' response to Reviewer #3's Comment #3

(1) PE and CE for devices having scattering layers with $S = 7.1$ vs $S = 11.5$

We thank Reviewer#3 for his or her kind attention to the details. As pointed out, EQE and PE values of OLEDs (in a geometry of Type A) with $S = 11.5$ are 50.6% and 198.4 lm/W while they are 50.9% and 196.8 lm/W with $S = 7.1$. That is, the latter combination has a slightly lower PE than the former even though its EQE is higher. The main reasons of such a seemingly contradictory trend are that (i) EQE and PE are presented at a given luminance value rather than at a given current density; and that (ii) the angular characteristics of the two devices are different. The difference in the angular characteristics is very small but non-negligible. It turned out devices with S of 7.1 have a slightly more side-angle component than those with S of 11.5 (See **Fig. R12**). In this case, a given luminance, measured at a direction normal to the substrate, can be realized with a smaller current density (J) in device with S of 11.5 than in devices with S of 7.1, if EQE are the same. Even though EQE is not exactly the same in this comparison, the difference in EQE is so small that the effect of the difference in the angular characteristics can make the aforementioned statement still valid. When a given luminance is obtained with lower J , its operating voltage will also be lower, making power efficiency larger, again if EQE values are the same or very similar between the two under comparison. The similar argument holds also for current efficiency (CE.)

Figure R12 | Dependence of the angular intensity of an OLED with SiO₂-based scattering layers with various *S* values.

To verify these notions, we summarized *J* and *V* of both devices at given luminance values in Table R2. One can easily see that the result clearly confirms our assertion. In short, such a seemingly unusual difference stems from to the fairly similar EQE values of the devices under comparison, a slight but still significant difference in the angular characteristics, and specifying the values at a given luminance than a given current density. Certainly, this does not represent an error in characterization.

Table R2 Comparison of J and V between the OLEDs with S =7.1 and 11.5			
L [cd m ⁻²]		Scatterance (S)	
		7.1	11.5
10	J [mA cm ⁻²]	6.0 × 10 ⁻³	5.8 × 10 ⁻³
	V [V]	2.45	2.43
1000	J [mA cm ⁻²]	0.63	0.61
	V [V]	4.09	4.06

(2) Regarding seemingly erratic current behavior below turn-on

Authors thank Reviewer#3 for his or her keen observations. “The inconsistent current density below the turn on” is due to the noise that occurred from the source-measure unit (Keithley 2400). The current density on the order of 10⁻⁵ to 10⁻⁶ mA/cm² corresponds to the actual current of 1 nA to 0.1nA considering the area of our devices (ca. 0.1 cm²). According to the specification of Keithley 2400, the noise in current measurement in its sensitivity range setting of 0 – 1 μA (= the highest sensitivity

range) and 0 –10 μA (the next highest sensitivity range) are 0.3 nA and 0.7 nA, respectively. In a real measurement environment involving interconnections through a glove box feedthrough, etc. the noise current can typically be worsening.

Below turn-on, the devices under study indeed showed J - V behavior limited by noise, exhibiting oscillatory current between positive and negative values, most of which were on the order of 0.01 nA to 0.1 nA with a few overshoots on the order of 0.1 to 1 nA. (The data points with negative current are missing in the graph as the present J - V curve were drawn in a log scale.) One can be assured that these noisy data are not used for calculation of device characteristics, since they appear below turn-on. After the turn-on, the current level is on the order of at least 50 nA to 500 nA or higher, which is large enough to overshadow the noise effect.

References for this response letter

- R1. Kim, K.-H., Moon, C.-K., Lee, J.-H., Kim, S.-Y. & Kim J.-J. Highly Efficient Organic Light-Emitting Diodes with Phosphorescent Emitters Having High Quantum Yield and Horizontal Orientation of Transition Dipole Moments. *Adv. Mater.* **26**, 3844-3847 (2014).
- R2. Kim, K.-H., Lee, S., Moon, C.-K., Kim, S.-Y., Park, Y.-S., Lee, J.-H., Lee, J. W., Huh, J., You, Y. & Kim, J.-J. Phosphorescent dye-based supramolecules for high-efficiency organic light-emitting diodes. *Nat. Commun.* **5**, 4769 (2014).

Table R3. Summary of the changes made in response to Reviewers' comment

Reviewer's comments	Changes made in response to reviewer's comments	Locations affected	
R1	1	 Title has been changed to reflect to the request regarding the importance of both scattering and horizontal dipole orientation in achieving the high efficiency. Experimental and calculation results made with other well-known emitters ($\text{Ir}(\text{ppy})_3$ and $\text{Ir}(\text{ppy})_2\text{acac}$) have been added to Supplementary Information (SI), and the corresponding description has been added to the main text. 	Title P12. Line 2 – 6 Supp. Fig. 8 Supp. Fig. 9 P16. Line 24-29
	2	 Figure 2 has been converted to contour plots showing the effect of both n_{SP} and d_{SP} to asymmetry parameter (g) as well as scattering efficiency (Q_{sc}). Implication of Q_{sc} has been discussed in terms of the upper limit for scattering coefficient (μ_{sc}) and the lower limit for the mean-free path (L_{MFP}). “Design ...” section has been revised to include discussion on how to choose (n_{SP} and d_{SP}) with a particular emphasis on the effect of Δn between scatterers and host medium. To compare with previous scattering-based approaches, TiO_2 ($n_{\text{SP}} = 2.5$, $d_{\text{SP}} = 200$ nm) based scattering layers were prepared and their characteristics have been shown in SI. Air void case was also mentioned in the main text and reference has been added. Figure 4 has been revised to include the experiments done with TiO_2 based scattering layers. It also includes new experimental results made with a large Al back reflector to mimic a situation for large-area devices. Discussion on this new addition has been included in the main text. A part of Fig.4 (on angular intensity/ spectral characteristics) has been separated as Fig. 5 to accommodate the change in Fig. 4. 	Fig. 2 P7. Line 2-4 P7. Line 5 -9 Supp. Fig. 4 P3. Line 8-11 P7. Line 10-20 Supp. Fig. 5. P8. Line 1-8 P15. Line 11-13 Fig. 4 P11. Line 4 -17 P11. Line 23 -29 Abstract P17. Line 12-14 Fig. 5
	3	Effect of Q_{sc} on the thickness requirement has been added in the context of Δn .	P7. Line 7-24 P8. Line 1-7
R2	1, 2	Please refer to the changes made for Reviewer #1's Comments #2.	
	3	The sentence of concern has been revised for clarification.	P4. Line 18
	4	The sentence of concern has been revised for clarification.	P16. Line 24
	5	The vertical axis of Fig. 1b has been changed to the total active layer thickness. Discussion and supplementary plots on its influence have been added to the main text and SI, respectively.	Fig. 1b P4. Line 3-9 P4. Line 20-24 Supp. Fig. 3
	6	Discussion and supplementary plot on the spectral dependence have been added to the main text and SI, respectively	P13. Line 6-11 Supp. Fig. 10 P15. Line 25-26
	8	The actual thickness of the scattering layers has been indicated in the Method Section and the SEM cross sectional images have been added.	P15. Line 18-19 Supp. Fig. 11
R3	1, 3	NA	

2	Please refer to the changes made for Reviewer #1's Comments #2	
---	--	--

REVIEWERS' COMMENTS:

Reviewer #2 (Remarks to the Author):

In the current revision NCOMMS-17-33001A, the authors have carefully and successfully addressed all points raised during the initial review and I have no further points of criticism. Hereby, I recommend this manuscript to be accepted at Nature Communications.

Reviewer #3 (Remarks to the Author):

The responses provided by the authors help clarify things, and the revised manuscript has been significantly improved. The experimental results on using the SiO₂ scatters do not represent major enhancement of the EQE. As a matter of fact, in the Type A devices, the champion efficiency is from the use of TiO₂ scatters. There are publications showing around 50% EQE without the use of horizontally oriented emitter. The originality and technical advancement are not sufficient for a Nature journal. However, the systematic simulation and analysis does provide a valuable contribution to methodology.

The title and focus of the work is external scattering. Results on the use of isotropic emitters should be added in the paper to examine the true experimental results from the optimized external scattering.

Response to Referees' Comments

Response to Reviewers' Comments

We thank the reviewers for their thoughtful comments and suggestions for improvements. Responses to each of the comments are summarized as follows:

Reviewer #3's comments:

Comment #1

The responses provided by the authors help clarify things, and the revised manuscript has been significantly improved. The experimental results on using the SiO₂ scatters do not represent major enhancement of the EQE. As a matter of fact, in the Type A devices, the champion efficiency is from the use of TiO₂ scatters. There are publications showing around 50% EQE without the use of horizontally oriented emitter. The originality and technical advancement are not sufficient for a Nature journal. However, the systematic simulation and analysis does provide a valuable contribution to methodology.

Authors' response to Reviewer #3's Comment #1

There are indeed reports showing single-junction OLEDs with around 50% EQE. However, all these results are for OLEDs based on a macroscopic half-ball lens or on a combination of internal and external light extraction structures. The former is for demonstration only and cannot be scaled up without undermining OLED's advantage as a planar, areal light source. The latter may provide a long-term solution and thus should continue to be developed, but there are many hurdles to overcome to implement internal outcoupling structures in a large scale and/or ensure stable operation without electrical short or local high field; therefore, it is vital, from the practical perspectives, to establish a systematic method to achieve highly efficient OLEDs based solely on the external light extraction schemes, in particular, those which are readily scalable at low cost. This is what the present study is all about. Among the reports on OLEDs adopting scalable external light extraction technologies without additional internal light extraction schemes, the present work does define the state of the art.

In addition, Reviewer#3 also claims that scattering layers based on SiO₂ are not better than those based on TiO₂. As mentioned in the 1st revised manuscript, there is no clear winner that can cover all the general cases; so it is a matter of choice that has to be made depending on the constraints of applications. For large-area devices that can be all right with a scattering layer of a few hundred micrometers, which is the case for the mainstream lighting application, SiO₂-based scattering layers will be a smarter choice because highest efficiency can be achieved over a wide range of scatterance (S), allowing for larger fabrication margin and improved panel-to-panel uniformity in brightness; on the other hand, if a target application requires ultrathin form-factors and thus desires scattering layers to be thin (e.g. a few tens of micrometers), TiO₂-based solutions will be a better choice as the EQE maximum occurs at low S .

Changes made in response to Reviewer #3's Comment #1

1) Sentences in Introduction have been revised so that the motivation and main scope of the present work can be more clearly identified.

Page 2

... Although combining both internal and external outcoupling schemes may result in the highest possible efficiency, internal outcoupling schemes can often be subject to electrical short, local high-field-induced

degradation, limited scalability of manufacturing, and/or increased fabrication cost. Likewise, although a half-ball lens has often been used as an external outcoupling structure and proven so effective in light extraction that OLEDs with EQE of 50% or higher can be realized with it^{13,16}, it is intended mainly for demonstration, rather than for any practical purpose; its use is limited to small-scale devices from the practical point of view and, furthermore, contradicts with a key benefit of an OLED as a planar or flexible light source. Therefore, it would be highly beneficial to develop a systematic way to significantly enhance outcoupling efficiency solely with scalable external outcoupling structures, such as microlens array foil, substrate texturing, or bulk-scattering film.

2) References have been added for the recent works reporting relatively high EQE values that we missed in the initial submission. The reference numbers have been updated to accommodate this addition.

14. Qu, Y., Kim, J., Coburn, C. & Forrest, S. R. Efficient, Nonintrusive Outcoupling in Organic Light Emitting Devices Using Embedded Microlens Arrays. *ACS Photonics* **5**, 2453-2458 (2018).

15. Ou, Q-D., Zhou, L., Li, Y-Q., Shen, S., Chen, J-D., Li, C., Wang, Q-K., Lee, S-T. & Tang, J-X. Extremely Efficient White Organic Light-Emitting Diodes for General Lighting. *Adv. Funct. Mater.* **24**, 7249-7256 (2014).

Comment #2

The title and focus of the work is external scattering. Results on the use of isotropic emitters should be added in the paper to examine the true experimental results from the optimized external scattering.

Authors' response to Reviewer #3's Comment #2

We thank Reviewer#3 for his or her thoughtful advice. We agree with Reviewer#3 that we should clarify how much enhancement results solely from optimized external scattering and from its combination with horizontal dipole orientation. In fact, we had included, in the 1st revision, the experimental results from OLEDs with the emitter of Ir(ppy)₃, which has isotropic dipole orientation. Those experimental data were also obtained with the full global optimization proposed in this work; however, with rather limited space, the description was rather short and didn't explicitly indicate that Ir(ppy)₃ has a random, isotropic orientation.

In this round of revision, the manuscript has been revised as follows so that this issue can be better clarified.

Changes made in response to Reviewer #3's Comment #2

1) Description has been added on the respective and combined contributions of horizontal dipole orientation and the proposed optimized external scattering layer to the EQE enhancement.

Page 4

In the proposed device configuration, the trans-scale simulation shows that $\eta_{\text{EQE}}^{(\text{max})}$ can reach 56.2 % when S , d_{ITO} , d_{ETL} , and d_{HTL} are 8.2, 150 nm, 65 nm, and 130 nm, respectively, with g approaching unity. In this active layer configuration, an OLED without a scattering layer would have η_{EQE} of ca. 28 % (Supplementary Fig. 3a, b). If an OLED without a scattering layer took the same configuration, but the emitter is randomly oriented ($\Theta = 0.67$), η_{EQE} would be limited at ca. 23 % (Supplementary Fig. 3c, d). The horizontal dipole orientation and the optimized scattering layer in the OLED under study are thus responsible for the EQE enhancement ratio of 1.2 (= 28/23) and 2.0 (= 56/28), respectively. The ratio of combined enhancement is therefore 2.4 (= 1.2×2.0) when compared with the EQE of the scattering-layer-free OLED based on an emitter having the same emission spectra and q as Ir(dmp₂ppy-ph)₂tmd but having Θ of 0.67.

Two important aspects can be noted from the overall trend in **Fig. 1b**. First, the dependence of $\eta_{\text{EQE}}^{(\text{max})}$ on d_{active} for a given set of g and S is relatively mild but still significant; therefore, it is important

2) As supplementary information for the description shown in 1), calculation results have been added as Supplementary Fig. 3c and 3d for an OLED with an emitter having the same emission spectra and q as $\text{Ir}(\text{dmpy-ph})_2\text{tmd}$ but having Θ of 0.67.

Supplementary Figure 3

Supplementary Figure 3 | The effect of total active layer thickness ($d_{\text{active}} = d_{\text{ITO}} + d_{\text{org.}}$) on the power fraction. (a-b) Power dissipation spectra of the reference device obtained for (a) $d_{\text{ITO}} = 80$ nm and (b) $d_{\text{ITO}} = 150$ nm. Here, d_x denotes the thickness of layer 'x' while 'org.' ITO, ETL, and HTL refer to organic, indium tin oxide, electron and hole transport layers, respectively. The case shown in (a) contains the global maximum air mode condition ($d_{\text{active}} = 245$ nm with $d_{\text{ITO}} = 80$ nm) with $d_{\text{ETL}} = 65$ nm without a scattering layer, and the case shown in (b) contains the condition maximizing the sum of air and substrate modes ($d_{\text{active}} = 385$ nm with $d_{\text{ITO}} = 150$ nm). The latter is found to correspond to the condition that eventually leads to the global maximum when the proposed scattering layer is used together. All the results in (a) and (b) are for OLEDs based on $\text{Ir}(\text{dmpy-ph})_2\text{tmd}$, which has horizontal dipole ratio (Θ) of 0.865. Power dissipation spectra of the reference device for (c) $d_{\text{ITO}} = 80$ nm and (d) $d_{\text{ITO}} = 150$ nm are presented also for an emitter same as $\text{Ir}(\text{dmpy-ph})_2\text{tmd}$ but having Θ of 0.67 (i.e. random, isotropic dipole orientation).

3) For experimental results obtained with $\text{Ir}(\text{ppy})_2\text{acac}$ and $\text{Ir}(\text{ppy})_3$, we have explicitly indicated that $\text{Ir}(\text{ppy})_2\text{acac}$ is an emitter with a slight preference toward horizontal orientation and that $\text{Ir}(\text{ppy})_3$ is an emitter with random, isotropic orientation.

● Page 8

... In a comparison experiment done with $\text{Ir}(\text{ppy})_2\text{acac}$ and $\text{Ir}(\text{ppy})_3$ - widely studied phosphorescent emitters with Θ of 0.76 (i.e. a slight preference toward horizontal orientation) and 0.67 (i.e. random, isotropic orientation)³³, respectively, optimized OLED devices (Type A) coupled with the SiO_2 -based scattering layers exhibited ...

- **Supplementary Figure 8 Caption**

Supplementary Figure 8 | Characteristics of Ir(ppy)₂acac-based organic light-emitting diodes with SiO₂ scattering layers. (a) External quantum efficiency (EQE) as a function of scatterance (S) (b) Current density (J) - luminance (L) - voltage (V) characteristics. (c-d) (c) EQE and (d) Power efficiency versus L. Results shown here are for Type A devices having optimized structure: ITO (150 nm) / TAPC (90 nm) / TCTA (10 nm) / TCTA:B3PYMPM:Ir(ppy)₂acac (8 wt.%, 30 nm) / B3PYMPM (65 nm) / LiF (1 nm) / Al (100 nm). Note that Ir(ppy)₂acac has a slight preference toward horizontal dipole orientation (Θ) being 0.76.

- **Supplementary Figure 9 Caption**

Supplementary Figure 9 | Characteristics of Ir(ppy)₃-based organic light-emitting diodes with SiO₂ scattering layers. (a) External quantum efficiency (EQE) as a function of scatterance (S) (b) Current density (J) - luminance (L) - voltage (V) characteristics. (c-d) (c) EQE and (d) Power efficiency versus L. Results shown here are for Type A devices having optimized structure: ITO (150 nm) / TAPC (80 nm) / TCTA (10 nm) / TCTA:B3PYMPM:Ir(ppy)₃ (8 wt.%, 30 nm) / B3PYMPM (65 nm) / LiF (1 nm) / Al (100 nm). Note that Ir(ppy)₃ has a random, isotropic dipole orientation (horizontal dipole orientation (Θ) = 0.67).

Editorial Comment:

* Your paper will be accompanied by a two-sentence editor's summary, of between 250-300 characters, when it is published on our homepage. Could you please approve the draft summary below or provide us with a suitably edited version.

Our response to the editorial comment:

We thank Editor very much for drafting a succinct summary of our work. Let us slightly revise it for clarification. You may choose one between Revision 1 and Revision 2.

(Original)

Current approaches to light extraction in organic light-emitting diodes (OLEDs) enable performance competitive to the state-of-the-art, but compromise on the technology's key benefits. Here, the authors demonstrate ultrahigh efficiency OLEDs via a device strategy based on forward light scattering.

(Revision 1) (318 characters including blanks)

Approaches based on a macroscopic lens to light extraction in organic light-emitting diodes (OLEDs) enable performance competitive to the state-of-the-art, but compromise on the technology's key benefits. Here, the authors demonstrate ultrahigh efficiency OLEDs via a device strategy based on forward light scattering.

(Revision 2) (315 characters including blanks)

Light extraction approaches based on a macroscopic lens in organic light-emitting diodes (OLEDs) enable performance competitive to the state-of-the-art, but compromise on the technology's key benefits. Here, the authors demonstrate ultrahigh efficiency OLEDs via a device strategy based on forward light scattering.

Corrections made to follow the journal policies and format requirements (Response to Editor's Comments)

We sincerely appreciate the editor for his or her valuable comments and suggestions for improvements. Responses to each of the comments are summarized as follows:

Reviewer #3's comments:

- Manuscript

● Page 1 (Title)

- The title must be less than 15 words and should not include punctuation
- : We reduced the length of the title and excluded punctuation as follows:

(15 words)

Lensfree OLEDs with over 50% external quantum efficiency via **tailored** external scattering and horizontally oriented **dipole** emitters

● Page 1 (Abstract)

- The abstract is too long. Please reduce to 150 words or less.
- Acronyms are not allowed in the Abstract. Please write the equivalent extended definition instead of the acronym.
- : We reduced the number of words and replaced all acronyms in abstract with the equivalent extended definition as follows:

(145 words)

High efficiency is important for successful deployment of any light sources. Continued efforts have recently made it possible to demonstrate **organic light-emitting diodes** with efficiency comparable to that of inorganic **light-emitting diodes**. However, such achievements were possible only with the help of a macroscopic lens or complex internal nanostructures, both of which undermine the key benefits of **organic light-emitting diodes** as an affordable planar light source. Here we present a systematic way to achieve **organic light-emitting diodes** with ultrahigh efficiency even only with an external scattering film, one of the simplest low-cost outcoupling structures. Through a global, multi-variable **high-speed** analysis, we show that scattering with a high degree of forwardness has a potential to play a critical role in realizing **OLEDs** with ultimate efficiency. Combined with horizontally oriented emitters, **organic light-emitting diodes** equipped with particle-embedded films tailored for forward-intensive scattering achieve a maximum external quantum efficiency of 56% **and a power efficiency of 221 lm W⁻¹**.

● Page 2

- Please refer to this section as Introduction
- : Section title of introduction paragraph has been added.

- Please make sure the main text is up to ~5000 words
- : Total number of words in the main text is less than 4300.

● Page 3 (Introduction)

-The results should be briefly described in the last paragraph of the introduction. Only the final paragraph of the introduction can discuss your results. To make this clear, please start it by, for example 'Here,' or 'We show', etc
- Please consider revising this paragraph or adding an additional paragraph that summarizes the results of the work.

: We revised previous introduction as follows:

-Please note that our style does not allow for the use of bold or italic for emphasis. Please remove this wherever it is used.

: Italic font in the text is edited.

(original)

... This equation-based character of the RTT makes it easy to combine, in a trans-scale fashion, with an optical model for light emission within an OLED, which may be generalized as radiative emission from a dipole in a thin-film multilayer stack. This then enables global, multi-variable analysis at high speed, in which an OLED with an external scattering layer (SL) is considered as a *whole*²³, rather than as two separate entities, so the design parameters of the scattering layer can be determined for maximal EQE in conjunction with those of the OLED layer configuration. Combined with highly oriented dipole emitters, SLs optimized in this way are shown to be able to yield OLEDs having EQE greater than 50% even without use of a macroscopic lens or complex internal nanostructure. Tailoring the characteristics of scattering layers such as asymmetry parameter, scattering efficiency, and scatterance is shown to play a key role in utilizing the full potential offered by the external scattering methods.

(Revised)

... This equation-based character of the RTT makes it easy to combine, in a trans-scale fashion, with an optical model for light emission within an OLED, which may be generalized as radiative emission from a dipole in a thin-film multilayer stack. This then enables global, multi-variable analysis at high speed, in which an OLED with an external scattering layer (~~SL~~) is considered as a **whole**²³, rather than as two separate entities, so the design parameters of the scattering layer can be determined for maximal EQE in conjunction with those of the OLED layer configuration.

With this approach, we here demonstrate that optimized external scattering layers, together with highly oriented dipole emitters, are able to yield OLEDs having EQE greater than 50% even without use of a macroscopic lens or complex internal nanostructure. Tailoring the characteristics of particle-embedded scattering layers such as asymmetry parameter, scattering efficiency, and scatterance is shown to play a key role in utilizing the full potential offered by the external scattering methods.

● Page 3

... schematically presented in **Fig. 1a (left)**. First, the radiant intensity ...

... wavelength of λ and angle θ exiting into a scattering layer (SL) [$= I_{\text{SL}}(\theta, \lambda)$] is calculated using the coherent dipole radiation theory ...

● Page 3 (subheading)

(Revised, 59 characters)

Global, ~~multi-variable~~ analysis of an OLED with an external scattering film.

● Page 4

... with the highest possible EQE are presented as a function of active layer thickness, which is the sum of indium tin oxide (ITO) and organic layer thicknesses ($d_{\text{active}} \equiv d_{\text{ITO}} + d_{\text{org}}$), of the OLED and g and S ...

... the optical constants of the layers used in this work are shown in **Fig. 1a (right)** and **Supplementary Fig. 2**, respectively. ...

- **Page 5 - 6 (subheading)**

The subsection previously entitled “**Design and characterization of forward-intensive scattering films and OLEDs made thereof**” has been divided into two subsections with shorter titles as summarized below:

(Original)

Design and characterization of forward-intensive scattering films and OLEDs made thereof. To investigate the condition for high g , therefore, ... Similarly, air voids ($n=1$)¹⁷ may also serve as useful scatterers; calculation indicates that g of up to ca. 0.8 and Q_{sc} approximately 1.8 can be readily available when used with the host medium in this work ($\Delta n = 0.57$) ... with a system having a relatively large Δn as shown in **Fig. 2a**.

Figure 3 presents the optoelectrical characteristics of the devices fabricated with the SiO₂-embedded scattering films with the thicknesses of the ITO and organic layers set at the optimal values obtained for the global maximum EQE. As an expected benefit of the external outcoupling scheme ...

(Revised)

Design and characterization of scattering films. To investigate the condition for high g , therefore, ... Similarly, air voids ($n=1$)¹⁹ may also serve as useful scatterers; calculation indicates that g of up to ca. 0.8 and Q_{sc} approximately 1.8 can be readily available when used with the host medium in this work ($\Delta n = 0.57$) ... with a system having a relatively large Δn as shown in **Fig. 2a**.

Performance and analysis of fabricated OLEDs. **Figure 3** presents the optoelectrical characteristics of the devices fabricated with the SiO₂-embedded scattering films with the thicknesses of the ITO and organic layers set at the optimal values obtained for the global maximum EQE. As an expected benefit of the external outcoupling scheme ...

- **Page 6**

... To vary S , concentrations of 0.2, 0.5, 1, 1.5, 3, and 4.5 wt.% were tried with d_{SL} fixed at approximately 290 μm (see method section). A scanning electron microscope image of a fabricated SiO₂-based scattering layer as well as a representative photograph of a scattering layer obtained from 3 wt.% solution is shown in Fig. 2c, d. The angular intensities of the light transmitted through these scattering films shown in **Fig. 2e** indicate that the measured data can be fitted well over a wide range of particle concentrations with ...

- **Page 7**

... even near the turn-on voltage (= ca. 2.4 V), but also current density - voltage curves are highly reproducible (see **Fig. 3a**). This allows for reliable measurement of EQE and power efficiency even at low brightness levels, ...

... which corresponded to a power efficiency of 197 lm W^{-1} and a current efficiency of 167 cd A^{-1} . Photographs of SiO₂-based OLEDs operating at current density of 0.1 mA cm^{-2} are shown in Fig. 3e.

To confirm the reliability of the proposed optical trans-scale simulation, ...

- **Page 8**

... exhibit color-stable performance regardless of the observation angle with ideal Lambertian-like angular characteristics (see Supplementary Fig. 10).

While the analysis and experiment in this work were done ...

- **Page 9**

-We prefer to avoid the use of terms like "new" and "novel" or "first", as in practice it is difficult to truly ascribe something as completely new. As such, it can detract from the achievements of the work by generating discussions about its "newness" instead of its unique aspects.

: We modified the sentence to avoid the term "first" as follows:

... EQE of 56% and a power efficiency of 221 lm W⁻¹, ~~which was the first demonstration done. This result is significant achievement in that such high efficiency is demonstrated~~ without a half-ball lens or a complex internal outcoupling scheme. ...

- **Page 10 (subheading in Methods)**

Measurement of the angular intensity of ~~transmitted~~ scattered light. After at least 30 mins of stabilization, ...

- **Page 10**

- We do not allow the use of speech marks for emphasized or approximations. Please remove the speech marks. We reserve the use of double quotation marks only for reported speech and quotations of other people's special usage of words. Single quotes may be used for non-attributed special uses, to replace 'so-called' or 'known as'.

: Speech marks are removed.

... and coherent dipole radiation theory also termed as "power-dissipation model"²² for OLED stacks^{24,25,34}. The latter takes into account Purcell effect, ...

- **Page 12**

Data availability. The authors declare that the data supporting the findings of this study are available within the article and its Supplementary Information files. Numerical values of data shown as graphs are available ~~upon request from the corresponding authors~~ from the corresponding authors upon reasonable request.

- **Page 15**

- Please refer to the competing financial interest statement as 'Competing Interests'

: Both 'Competing financial interests' are replaced with 'Competing interests'.

Competing ~~financial~~ interests: The authors declare no competing ~~financial~~ interests.

- **Figures 1**

- Please avoid using just colours to identify features in a figure - this might become confusing if a reader prints the paper in black & white.

: Line styles of **Fig. 1d, e** are divided as follows:

-Panels in figures should be labelled using the a, b, c... convention and should not contain I, II, IV, A, B, Z, Top, Left, Bottom etc.
 : (Left) and (Right) labels are deleted.

-Ensure all abbreviations are defined in the figure caption.
 : We modified the figure caption to define all abbreviations as follows:

Figure 1 | Schematic diagram of the proposed trans-scale optical simulation and global optimization results. (a) (Left) Schematic illustration of suggested trans-scale optical simulation—(right) and schematic diagram of the detailed **organic light-emitting diode (OLED)** structure under study as well as parameters used in multi-variable analysis. (b) Three-dimensional slice plot of maximum achievable **external quantum efficiency (EQE)** [$= \eta_{EQE}^{(max)}$], vs. **active layer thickness** ($d_{active} = d_{ITO} + d_{org.}$), **asymmetry parameter** (g), and **scatterance** (S). d_{ITO} , d_{ETL} , and d_{HTL} were varied to find $\eta_{EQE}^{(max)}$ for each $\{d_{active}, g, S\}$. Here, d_x denotes the thickness of layer 'x' while 'org.', ITO, ETL, and HTL refer to organic, indium tin oxide, electron and hole transport layers, respectively. The global maximum considering all the parameters is represented by a black star. (c) ...

● **Figures 2**

- As we are not as restrained for space as print journals, to reduce figure clutter and improve readability, please enlarge the inset into a full panel of its own, so that it is clearer for the reader.
 - Please enlarge the inset in Figure 2c as well.
 : Enlarged inset images of **Fig. 2a, b** are added to **Fig. 2** as **Fig. 2c** and **Fig. 2d**.

- Ensure all abbreviations are defined in the figure caption.
- : We modified the figure caption to define all abbreviations as follows:

Figure 2 | Design and analysis of a scattering layer. (a-b) Calculated (a) asymmetry parameter (g) and (b) logarithm of the scattering efficiency (Q_{sc}) of a scattering layer at 532 nm according to the diameter (d_{sp}) and refractive index (n_{sp}) of a scattering particle (SP) in the host (NOA 73, $n_{SL} \sim 1.57$). ~~Inset in (a): An SEM image of a fabricated SiO₂-based scattering layer. (Scale bar: 5 μ m) (c) Measured (dot) and fitted (dashed) graphs of angular characteristic of scattered light after passing through SiO₂-embedded scattering layers. 532 nm laser was used for the measurement. Inset: Photographs of glass without a scattering layer and with a scattering layer fabricated using 3 wt.% solution. (e) A scanning electron microscope image of a fabricated SiO₂-based scattering layer. (Scale bar: 5 μ m) (d) Photographs of glass without a scattering layer and with a scattering layer (SL) fabricated using 3 wt.% solution. (e) Measured (dot) and fitted (dashed) graphs of angular characteristic of scattered light after passing through SiO₂-embedded scattering layers. 532 nm-laser was used for the measurement.~~

● Figures 3

- Please avoid using just colours to identify features in a figure - this might become confusing if a reader prints the paper in black & white.
- : Please feel free to let us know if it is not fine to just use different symbols as the previous version of the figure does.

- Ensure all abbreviations are defined in the figure caption.
- : We modified the figure caption to define all abbreviations as follows:

Figure 3 | Device Characteristics of fabricated organic light-emitting diodes (OLEDs) with a SiO₂-based scattering layers having various scatterance (S). (a) Current density (J) - luminance (L) - voltage (V) characteristics. (b-d) External quantum efficiency, (b) Power efficiency, and (c) Current efficiency (c) versus L . Layout of the devices used in this figure follows that of Type A defined in Fig. 4. (e) Photographs of OLEDs with SiO₂-embedded scattering layers according to scatterance (S). Photos were taken at current density of 0.1 mA cm⁻².

● Figures 4

- As we are not as restrained for space as print journals, to reduce figure clutter and improve readability, please enlarge the inset into a full panel of its own, so that it is clearer for the reader.
- : Enlarged inset images of **Fig. 4a** are added to **Fig. 3** as **Fig. 3e**.

- Ensure all abbreviations are defined in the figure caption.
 : We modified the figure caption to define all abbreviations as follows:

Figure 4 | The performance of organic light-emitting diodes (OLEDs) with the proposed scattering layers including devices fabricated on large substrates. (a) The measured external quantum efficiencies (EQEs) vs. scattering (S) of OLEDs with SiO₂-based (asymmetry parameter (g) = 0.91) and TiO₂-based (g = 0.68) scattering layers. The color-matched dotted lines indicate LightTools simulation results that reflect the actual geometry (Type A) used for EQE- S trend study. ~~Inset: Photographs of SiO₂-based OLEDs according to S .~~ (b) Device test structures ... Photographs of OLEDs (Type B) are also shown for both SiO₂-based (S = 7.1) and TiO₂-based OLEDs (S = 2.0). (Scale bar: 5 mm) (c-e) Device characteristics of Type B champion OLEDs.

● **Figures 5**

- Please avoid using just colours to identify features in a figure - this might become confusing if a reader prints the paper in black & white.
 : Line styles of **Fig. 5a** are divided as follows:

- Please enlarge the inset into a full panel of its own, so that it is clearer for the reader.
 : We moved the inset of **Fig. 5a** to supplementary information as **Supplementary Fig. 10** to enlarge the inset while preserving the balance of the original figure (**Fig. 5**).

Figure 5 | Angular spectral dependence of organic light-emitting diodes (OLEDs) with the proposed SiO₂-based scattering layers. (a) Experimental (semi-transparent symbols) and simulated (dashed lines) normalized angular spectra of OLEDs with a 3 wt.% scattering layer (scattering (S) = 7.1). ~~Inset: Measured normalized angular EL intensity and comparison with the simulation results. Black dotted line represents the Lambertian distribution.~~ (b) CIE coordinates of angular spectrums of the devices with different SiO₂-based scattering layers.

- **Supplementary information**

● **Supplementary Figures 3**

- The figure title should be no longer than one line of a word document. Please shorten the figure title.
 : The title is modified to fit within on line.

Supplementary Figure 3 | The effect of total active layer thickness ($d_{\text{active}} = d_{\text{ITO}} + d_{\text{org.}}$) on the power fraction delivered into substrate. (a-b) Power dissipation spectra of the reference device obtained for (a) $d_{\text{ITO}} = 80$ nm and (b) $d_{\text{ITO}} = 150$ nm. Here, d_x denotes the thickness of layer 'x' while 'org.', ITO, ETL, and HTL refer to organic, indium tin oxide, electron and hole transport layers, respectively. The case shown in (a) contains the global maximum air mode condition ...

● **Supplementary Figures 4**

Supplementary Figure 4 | Definition of scattering efficiency and its relation to key scattering parameters (a) The definition of scattering efficiency (Q_{sc}) and its relation to (i) the maximum scattering coefficient (μ_{sc}) achievable for simple cubic arrangement of nanoparticles ($\mu_{\text{sc}}^{(\text{c,max})}$) and (ii) the corresponding minimum mean-free path (MFP) [= L_{MFP}] for the same particle arrangement ($L_{\text{MFP}}^{(\text{c,min})}$). (b) Logarithm value of ...

● **Supplementary Figures 5**

- As we are not as restrained for space as print journals, to reduce figure clutter and improve readability, please enlarge the inset into a full panel of its own, so that it is clearer for the reader.
 : Since we think that the inset figure does not give meaningful information to the reader, we decide to omit the inset figure to preserve the balance of the original figure

- Please avoid using just colours to identify features in a figure - this might become confusing if a reader prints the paper in black & white.
 : Line styles of **Supplementary Fig. 5b** are divided as follows:

Supplementary Figure 5 | TiO₂-based Scattering layers prepared by dispersing TiO₂ nanoparticles (diameter = 200 nm) dispersed in the NOA 73 host. (a) Measured (dot) and fitted (dashed) graphs of angular characteristic of scattered light after passing through the TiO₂-embedded scattering layers with different level of concentration in wt.%. From the best fit, asymmetry parameter (g) is found to be 0.68. Inset: A SEM image of a fabricated TiO₂-based scattering layer. (Scale bar: 1 μm) (b) The comparison between ...

● **Supplementary Figures 6**

- Please shorten the figure title to one line of a word document in length.
 : The title is modified to fit within on line.

- Please enlarge the insets in Supplementary Figure 6b into their own full panels for clarity.
 : Enlarged inset images of **Supplementary Fig. 2b** are added as **Supplementary Fig. 2c** and **Supplementary Fig. 2d**.

- Ensure all abbreviations are defined in the figure caption.
 : We modified the figure caption to define all abbreviations as follows:

Supplementary Figure 6 | Configurations of a device under study used in LightTools™ simulation done to see the importance of a back reflector in matching the experimental data and the idealized simulation. (a) Oblique top view of the geometry (type A in the main text) used in the LightTools™ simulation as well as in the experiment. The orange area is the device active region where emission occurs, and a surrounding gray area illustrates the aluminum reflector, which we placed below the cathode of an **organic light-emitting diode (OLED)** to reduce the finite size effect, as described in Method Section. **(b)** Side view of **the simulation geometries and simulation** ray diagrams comparing without (top) and with the reflector (bottom). The simulation results clearly

show that a significant portion of the emitted rays proceeds backward by back scattering. Without the reflector, those rays would pass through the metal-free portion, exiting the device without returning to the scattering layer. ~~The inset photo, taken from the rear side of an OLED with the scattering layer, clearly illustrates such optical loss.~~ With a reflector, such loss is suppressed to a significant degree although a small portion can exit the device in the rear side, unless the reflector. The dimensions in (a) and (b) are for the configuration defined as Type A in Fig. 4 of the main text. ~~(c) A photograph taken from the rear side of an OLED with a scattering layer. This photo clearly illustrates that nontrivial amount of light propagates into backward.~~ ~~(d) Side view of the simulation geometry.~~

● Supplementary Figures 7

- Ensure all abbreviations are defined in the figure caption.
: We modified the figure caption to define all abbreviations as follows:

Supplementary Figure 7 | Device Characteristics of $\text{Ir}(\text{dmpvy-ph})_2\text{tmd}$ -based organic light-emitting diodes with a TiO_2 -based scattering layers having various S . (a) External quantum efficiency (EQE) as a function of scatterance (S) (b) Current density (J) - luminance (L) - voltage (V) characteristics. ~~(c-d)~~ (c) EQE and (d) Power efficiency versus L

● Supplementary Figures 8

- Please shorten figure title
: The title is modified to fit within on line.

- Ensure all abbreviations are defined in the figure caption.
: We modified the figure caption to define all abbreviations as follows:

Supplementary Figure 8 | Device Characteristics of $\text{Ir}(\text{ppy})_2\text{acac}$ -based organic light-emitting diodes based on the emitter of $\text{Ir}(\text{ppy})_2\text{acac}$ with a SiO_2 -based scattering layers having various S . (a) External quantum efficiency (EQE) as a function of scatterance (S) (b) Current density (J) - luminance (L) - voltage (V) characteristics. ~~(c-d)~~ (c) EQE and (d) Power efficiency versus L

● Supplementary Figures 9

- Please shorten figure title and define all abbreviations in the figure caption.
: The title is modified to fit within on line. In addition, we modified the figure caption to define all abbreviations as follows:

Supplementary Figure 9 | Device Characteristics of $\text{Ir}(\text{ppy})_3$ -based organic light-emitting diodes based on the emitter of $\text{Ir}(\text{ppy})_3$ with a SiO_2 -based scattering layers having various S . (a) External quantum efficiency (EQE) as a function of scatterance (S) (b) Current density (J) - luminance (L) - voltage (V) characteristics. ~~(c-d)~~ (c) EQE and (d) Power efficiency versus L

● Supplementary Figures 10

Supplementary Figure 10 | Normalized angular electroluminescence (EL) intensity. Normalized angular EL intensity (red hollow boxes) of OLEDs with a SiO_2 scattering layer, whose scatterance is 7.1, is compared with simulation result (blue dashed line). Black dotted line represents the Lambertian distribution.

● Supplementary Figures 11

Supplementary Figure 11 | Asymmetry parameter vs. wavelength for the SiO_2 ($d_{\text{SP}} = 800 \text{ nm}$) and TiO_2 ($d_{\text{SP}} = 200 \text{ nm}$) particles embedded in NOA73 host.

- **Supplementary Figures 12**

(original)

Supplementary Figure 11 | Scanning electron microscopy images of the cross-sections of scattering layers. The average thickness measured from 12 positions is $287 (\pm 11) \mu\text{m}$.

(revised, bold font is edited)

Supplementary Figure 12 | Scanning electron microscopy images of the cross-sections of scattering layers. The average thickness measured from 12 positions is $287 (\pm 11) \mu\text{m}$.

- **Supplementary Figures 13**

Supplementary Figure 13 | Comparison between ~~the~~ simulation results from ~~custom-made simulation based on~~ radiative transfer theory (RTT) and ~~commercially available Monte-Carlo-based~~ ray-tracing simulation. (a) The simulation structure ...

Voluntary Correction

1) Clarification has been made for Supplementary Figure 7. (All the data presented in the main text are for Ir(dmppy-ph)₂tmd-based OLEDs, but those presented in Supplementary Information (SI) include data from Ir(ppy)₃ and Ir(ppy)₂acac; and thus the figure captions in SI should clearly indicate which devices the corresponding figures are for.)

Supplementary Figure 7 Caption

Supplementary Figure 7 | Characteristics of Ir(dmppy-ph)₂tmd-based organic light-emitting diodes with TiO₂ scattering layers. (a) External quantum efficiency (EQE) as a function of scatterance (*S*) **(b)** Current density (*J*) - luminance (*L*) - voltage (*V*) characteristics. **(c-d)** **(c)** EQE and **(d)** Power efficiency versus *L*. Results shown here are for **Type A devices having optimized structure: ITO (150 nm) / TAPC (130 nm) / TCTA (10 nm) / TCTA:B3PYMPM:Ir(dmppy-ph)₂tmd (4 wt.%, 30 nm) / B3PYMPM (65 nm) / LiF (1 nm) / Al (100 nm).**

2) Clarification has been made for Table 1.

- Title has been changed to specify which OLED devices Table 1 is about.

Table 1 | Summary of the performance of the OLEDs with SiO₂-based scattering layers having various *S*.

- The row of *S* = 7.1 : the results from Type-B OLEDs have also been included.

7.1	50.9 (56.3 [†])	48.1 (52.6)	196.8 (221.1)	111.6 (136.4)	167.0 (169.6)	158.1 (158.4)
------------	---------------------------	-------------	---------------	---------------	---------------	---------------

- Comments at the bottom of Table 1 : a clarifying comment has been added.

[†] The values inside parenthesis are for the Type-B OLED in Fig. 4. All the other data are from Type-A OLEDs.